# Detection of Malicious Websites Using Symbolic Classifier

**Nikola Anđelić** *,†, **Sandi Baressi Šegota**, **Ivan Lorencin** † and **Matko Glučina**

Faculty of Engineering, University of Rijeka, 51000 Rijeka, Croatia
* Correspondence: nandelic@riteh.hr
† These authors contributed equally to this work.

**Abstract:** Malicious websites are web locations that attempt to install malware, which is the general term for anything that will cause problems in computer operation, gather confidential information, or gain total control over the computer. In this paper, a novel approach is proposed which consists of the implementation of the genetic programming symbolic classifier (GPSC) algorithm on a publicly available dataset to obtain a simple symbolic expression (mathematical equation) which could detect malicious websites with high classification accuracy. Due to a large imbalance of classes in the initial dataset, several data sampling methods (random undersampling/oversampling, ADASYN, SMOTE, BorderlineSMOTE, and KmeansSMOTE) were used to balance the dataset classes. For this investigation, the hyperparameter search method was developed to find the combination of GPSC hyperparameters with which high classification accuracy could be achieved. The first investigation was conducted using GPSC with a random hyperparameter search method and each dataset variation was divided on a train and test dataset in a ratio of 70:30. To evaluate each symbolic expression, the performance of each symbolic expression was measured on the train and test dataset and the mean and standard deviation values of accuracy (ACC), $AUC$, precision, recall and f1-score were obtained. The second investigation was also conducted using GPSC with the random hyperparameter search method; however, 70%, i.e., the train dataset, was used to perform 5-fold cross-validation. If the mean accuracy, $AUC$, precision, recall, and f1-score values were above 0.97 then final training and testing (train/test 70:30) were performed with GPSC with the same randomly chosen hyperparameters used in a 5-fold cross-validation process and the final mean and standard deviation values of the aforementioned evaluation methods were obtained. In both investigations, the best symbolic expression was obtained in the case where the dataset balanced with the KMeansSMOTE method was used for training and testing. The best symbolic expression obtained using GPSC with the random hyperparameter search method and classic train–test procedure (70:30) on a dataset balanced with the KMeansSMOTE method achieved values of $\overline{ACC}$, $\overline{AUC}$, $\overline{Precsion}$, $\overline{Recall}$ and $\overline{F1\text{-}score}$ (with standard deviation) $0.9992 \pm 2.249 \times 10^{-5}$, $0.9995 \pm 9.945 \times 10^{-6}$, $0.9995 \pm 1.09 \times 10^{-5}$, $0.999 \pm 5.17 \times 10^{-5}$, $0.9992 \pm 5.17 \times 10^{-6}$, respectively. The best symbolic expression obtained using GPSC with a random hyperparameter search method and 5-fold cross-validation on a dataset balanced with the KMeansSMOTE method achieved values of $\overline{ACC}$, $\overline{AUC}$, $\overline{Precsion}$, $\overline{Recall}$ and $\overline{F1\text{-}score}$ (with standard deviation) $0.9994 \pm 1.13 \times 10^{-5}$, $0.9994 \pm 1.2 \times 10^{-5}$, $1.0 \pm 0$, $0.9988 \pm 2.4 \times 10^{-5}$, and $0.9994 \pm 1.2 \times 10^{-5}$, respectively.

**Keywords:** genetic programming; malicious websites; oversampling methods; symbolic classifier; undersampling methods

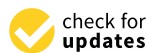

## 1. Introduction

A malicious website is a website that attempts to run any software that will disrupt computer operation, gather your personal information, or gain total control over your device. These types of sites are a common and serious threat to cybersecurity [1]. The malicious websites host unsolicited content such as spam, phishing, drive-by downloads, etc., which lure users into becoming victims of monetary loss, theft of private information,

and malware installation. The traditional way of handling malicious websites is done through the usage of blacklists which can be exhaustive and cannot detect newly generated malicious websites. In the last decade, the research is focused on the implementation of machine learning (ML) algorithms to improve the generality of malicious website detection.

Today, there are various approaches used to solve the problem of malicious website detection. These approaches are divided into two categories, i.e., blacklisting or heuristics and using ML algorithms.

### 1.1. Blacklisting or Heuristics Approach

The blacklisting or heuristics approaches are classical approaches used for detecting malicious websites. The blacklisting approaches are based on creating and maintaining a list of websites that are known to be malicious. When a new website is visited a database search is performed and if the website is stored in the blacklist the warning will be generated. Otherwise, it is assumed to be a benign website. The advantage of using a blacklist approach is their simplicity and efficiency so they are most commonly used by many anti-virus software's [2]. According to [3], the disadvantage of this approach is exhaustive list maintenance since new websites are generated daily so it is impossible to detect new threats. The heuristic approaches are blacklist methods extension and are based on signature blacklist development. In this case, the common attacks are identified and a signature is assigned to this attack type. The intrusion detection systems scan the web page for such signatures and if they exit the flag is raised. When compared to blacklist methods these methods have better generalization capabilities since it is possible to detect new threats on new websites. The disadvantage of these methods is that they are designed for a limited number of common threats and cannot be generalized to all types of newer attacks.

### 1.2. Machine Learning Algorithms

Machine learning algorithms try to analyze the information of the website and corresponding websites or web pages through the extraction of good feature representations that will be used for training a prediction model on training data of both malicious and benign websites. Two types of features can be used and these are static and dynamic features. According to [4–7], in the analysis of static features, the analysis of a web page is based on information available (mainly execution of JavaScript code) without execution of a web page. Usually, the extracted features are URL string, host information, HTML, and JavsScript code. Static methods are safer than dynamic methods since no execution is required however, the assumption is that the distribution of these features is different in the case of malicious websites. In a dynamic approach, systems which are potential victims are monitored to look for any anomalies. According to [8,9], this can be achieved using system calls to detect malicious JavaScript and harness the power of big data to detect malicious URLs based on known ones using the Internet access logs.

So far, various ML algorithms have been used for the detection of malicious websites such as support vector machines (SVM), Naive Bayes, detection trees, and others. The SVM is one of the most used supervised learning algorithms in the detection of malicious websites.

In [10], the authors used an SVM classifier to determine whether a web page is a legitimate or a phishing page. The results of the investigation showed that the proposed phishing detector can achieve a high accuracy rate with relatively low false positive and low false negative rates. The authors in [11] developed a system based on SVM that can detect malicious web pages using features extracted from dynamic HTML pages. The result of the investigation showed that this method is resilient to code obfuscations and can detect malicious websites with high accuracy. In [12], the authors presented a novel cross-layer malicious website detection approach that analyzes network-layer traffic and application-layer website contents simultaneously. The results of the investigation showed that cross-layer detection is 50 times faster than the dynamic approach while detection can be almost as effective as the dynamic approach.

In [13], the authors developed a reliable filter named Prophiler that uses static analysis to examine web-page for malicious content by taking into account the following features derived from HTML contents of the web page, associated JavaScript code, and corresponding URL. The derived features for millions of pages were used for training and testing Random Tree, Random Forest, Naive Bayes, J48, and Byes Net classifiers. The results showed that the developed filter is 85% effective than dynamic analysis. In [14], the dataset was developed which contains four feature types (page, domain, URL type, and word features) of several million URLs that were used to train and test logistic regression. The results showed that logistic regression can successfully detect phishing URLs. The misuse of the short URLs and characteristics of the spam and non-spam short URLs were analyzed in [15]. The dataset, i.e., short URLs, were collected from Twitter, and features were extracted and used for training/testing random trees, random forest, star, decision tree, decision table, logistic regression, and SVM classifier. The results showed that the random tree classifier algorithm achieved the best classification performance (accuracy 90.81%, F1 score 0.913). The detection of malicious URLs that were propagated using forwarding behavior on online social networks has been analyzed in [16] using Bayes Net, J48, and Random Forest classifier. The Bayes Net achieved the highest classification accuracy of almost 85%.

The detection of malicious short URLs on Twitter has been investigated in [17] using random forest and SVM classifier. The investigation showed that both algorithms could detect short malicious URLs from Twitter with high classification accuracy. The Extreme Learning Machines have been used in [18] to classify phishing websites. The proposed method showed promising results in the detection of phishing websites.

The convolutional (CNN) and recurrent neural networks (RNN) have been used in [19] to classify domain names as either benign vs. produced by malware. The dataset used in this research consisted of 2 million domain names and results showed that both CNN and RNN have high but similar classification accuracy.

Numerous research papers describe the successful implementation of various ML algorithms in the detection of malicious websites. So, for this paper, the chronological overview of ML algorithms implementation in the detection of malicious websites with results is summarized in Table 1.

As seen from Table 1, all research papers used accuracy as an evaluation method to measure the performance of used ML algorithms in the detection of malicious websites.

### 1.3. Definition of the Research Idea, Novelty, Research Hypotheses, and Scientific Contributions

The main disadvantage of the previously presented research papers is that, after the application of the ML algorithm, i.e., the training process, the model is obtained which can not be transferred into a simple symbolic expression (mathematical equation). This ML model can be saved and called when needed which requires additional computational resources.

The idea of this paper is to obtain a symbolic expression (simple mathematical equation) that can detect malicious websites with high classification accuracy using genetic programming symbolic classifier (GPSC). The advantage of using this approach is that the result is a symbolic expression (mathematical equation) that requires less disk space for storage and is easier to use (faster investigation) when exploring a new website.

Generally, genetic programming (GP), according to [20] is a technique of evolving a randomly generated initial population of symbolic expressions that are unfit for the particular task and throughout the evolution process by applying genetic operations (crossover and mutation) fit them for the particular task. The GP has some similarities with the genetic algorithm (GA) and these are initial population, genetic operations (crossover and mutation), fitness function, stopping criteria, etc. However, GP has some similarities with supervised ML algorithms since it requires a dataset provided with input and output variables.

**Table 1.** List of research papers in which different ML algorithms were used for the detection of malicious websites with achieved results.

| References | ML or Deep Learning Classifiers | Results |
|---|---|---|
| [14] | LR | ACC: 93.4% |
| [5] | Naïve Bayes, SVM, LR | ACC: 95–99% |
| [6] | MLP-Classifier, LR with SGD, PA, CW, | Acc. 99% |
| [11] | Decision Tree, Naïve Bayes, SVM, Boosted Decision Tree | ACC: 96% |
| [7] | Perceptron, LR with SGD, PA, CW | ACC: 99% |
| [10] | SVM | TP: 97.33%, FP: 1.45% |
| [4] | J48, Random Tree, Random forest, Naïve Bayes, Bayes Net, SVM, LR | ACC: 97% |
| [12] | Naïve Bayes, LR, SVM, J48 | ACC: 99.178%, FN: 2.284%; 0.422% |
| [15] | Random Tree, Random Forest, KStar, Decision Tree, Decision Table, Simple Logistic, SVM | ACC: 90.81%; F1-score: 91.3% |
| [8] | SVM | ACC: 96% |
| [16] | Bayes Net, J48, Random Forest | ACC: 84.74%; F1-score: 83% |
| [18] | ELM, BPNN, SVM, NB, k-NN, OPELM, Adaboost ELM, MV-ELM, LC-ELM | ACC: 9904% |
| [19] | CNN, RNN | ACC: 97–98% |

ACC—Accuracy, ELM—Extreme Learning Machine, PA—Passive Aggressive, MLP-Classifier—Multilayer Perceptron Classifier, Logistic Regression, SGD—Stochastic Gradient Descent, CW—Confidence–Weighted.

The novelty of this paper is to show the procedure of how GPSC can be utilized to obtain the symbolic expression for the detection of malicious websites. Besides the GPSC utilization, the dataset preparation procedure is presented, as well as dataset balancing methods since the original dataset has an imbalance between class samples. To obtain the symbolic expression with high classification accuracy the random hyperparameter search method and 5-fold cross-validation methods were developed and used in this research.

Based on the extensive literature overview, and the idea and novelty of this paper, the following questions arise:

- Is it possible to utilize the GPSC algorithm in combination with random hyperparameter search method to obtain simple symbolic expression which could be used for classification of malicious websites with high accuracy?
- Is it possible to utilize GPSC in combination with random hyper-parameter search and 5-fold cross-validation to improve the classification accuracy of malicious websites?
- Do dataset oversampling and undersampling methods have some influence on the classification accuracy of malicious websites?

The scientific contributions are:

- Investigate if GPSC can be applied to the dataset for the detection of malicious websites;
- Investigate if datasets balanced with undersampling and oversampling methods have any influence on classification accuracies of obtained symbolic expressions using GPSC algorithm;
- Investigate if the random hyperparameter search method, as well as 5-fold cross-validation, has any influence on the classification accuracy of obtained symbolic expressions in the detection of malicious websites.

The structure of the paper is divided into the following sections: Materials and Methods, Results, Discussion, and Conclusions. In Materials and Methods, the research methodology is presented as well as the dataset description and preparation, the GPSC algorithm, random hyper-parameter search, cross-validation, and the evaluation methodology. In results, the results of an extensive investigation are presented. In discussion, the obtained results are discussed and in conclusions, the conclusions are provided based on the discussion and hypothesis given in the introduction section.

## 2. Materials and Methods

In this section, the research methodology is presented as well as the used dataset (description and preparation), GPSC algorithm, random hyper-parameter search method, 5-fold cross-validation, and evaluation methodology.

### 2.1. Research Methodology

As already stated in the Introduction section, the idea of this paper is to detect malicious websites using the GPSC algorithm. To do this, the publicly available dataset [21–23] was used which must be transformed into a suitable format so the GPSC algorithm can be applied. Since the dataset has a large imbalance of classes, i.e., a large number of benign and a small number of malicious websites the idea was also to investigate if the application of undersampling and oversampling methods could balance the dataset classes and in the end improve the classification accuracy. After dataset transformation into a usable format, a total of seven different dataset variations were created including the original unbalanced dataset (after dataset transformation into usable format). The remaining six dataset balancing methods are:

- Random undersampling;
- Oversampling methods;
    - Random oversampling;
    - Adaptive Synthetic (ADASYN) method;
    - Synthetic Minority Oversampling TEchnique (SMOTE);
    - Borderline Synthetic Minority Oversampling TEchnique (BorderlineSMOTE), and
    - Application of KMeans clustering before oversampling using Synthetic Minority Oversampling Technique (KMeansSMOTE).

It should be noted that there are other oversampling/undersampling methods; however, they were omitted from further investigation since they did not balance the dataset. The procedure of training and testing the GPSC was performed on each dataset variation and can be divided into two main approaches i.e.,

- Train/test GP symbolic classifier with random hyper-parameter search method; and
- Train/test GP symbolic classifier with random hyper-parameters search and with 5-fold cross-validation on train dataset part.

The schematic overview of the research methodology is shown in Figure 1.

### 2.2. Dataset Description and Preparation

The publicly available dataset used in this research can be downloaded from [23]. As already stated, the dataset had to be transformed into a numeric format to be used in the GPSC algorithm. The initial form of the dataset is shown in Table 2.

In Table 2, the website classification dataset is presented in its original form. The dataset consists of url, url_len, ip_add, geo_loc, tld, who_is, https, js_len, js_obf_len, content, and the label. The url is a variable containing web addresses of benign and malicious websites and the variable name stands for Uniform Resource Locator, also known as web address [24]. The ur_len is the length of the url address, i.e., the number of characters in the web address. The ip_add is the internet protocol address of the website. The geo_loc is the country from which the website originates. The tld stands for a top-level domain (com, de, net, ...). The who_is variable describes if the website has complete or incomplete infor-

mation available at who_is website [25]. The https variable has two values "yes" and "no" which represent if the website has a hypertext transfer protocol secure or not. The js_len variable represents the length of JavaScript code on the webpage. The js_obf_len variable represents the length of the obfuscated JavaScript code. The content variable represents the raw webpage content including the JavaScript code. Finally, the label variable represents the class label for the benign or malicious website. All variables except the label variable will eventually end up as the input to the GP symbolic classifier and the label will be the output variable used for training and testing GP symbolic classifier. However, the main problem of the dataset is that majority of variables are of string type which means that some kind of data transformation is needed to represent all the variables in numeric format.

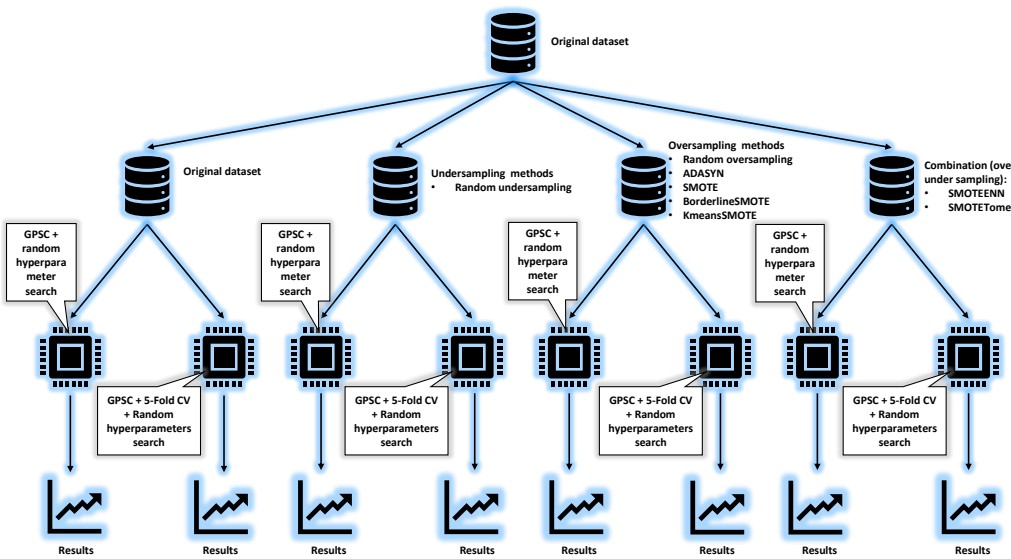

**Figure 1.** The schematic overview of research methodology.

**Table 2.** The example of the dataset in initial form (accessed on 10 October 2022).

| url | url_len | ip_add | geo_loc | tld | who_is | https | js_len | js_obf_len | Content | Label |
|---|---|---|---|---|---|---|---|---|---|---|
| https://members.tripod.com/russiastation/ | 40 | 42.77.221.155 | Taiwan | com | complete | yes | 58 | 0 | Named themselves charged ... | good |
| https://www.ddj.com/cpp/184403822 | 32 | 3.211.202.180 | United States | com | complete | yes | 52.5 | 0 | And filipino field .... | good |
| https://www.naef-usa.com/ | 24 | 24.232.54.41 | Argentina | com | complete | yes | 103.5 | 0 | Took in cognitivism, whose ... | good |
| http://www.ff-b2b.de/ | 21 | 147.22.38.45 | United States | de | incomplete | no | 720 | 532.8 | fire sodomize footaction tortur ... | bad |
| https://us.imdb.com/title/tt0176269/ | 35 | 205.30.239.85 | United States | com | complete | yes | 46.5 | 0 | Levant, also monsignor georges ... | good |
| https://efilmcritic.com/hbs.cgi?movie=311 | 40 | 8.28.167.23 | United States | com | complete | yes | 39.5 | 0 | Signals ... | good |
| https://christian.net/ | 21 | 125.223.123.231 | China | net | complete | yes | 136 | 0 | Temperature variations ... | good |
| https://www.indsource.com | 24 | 208.169.193.185 | United States | com | complete | yes | 51 | 0 | Were; an optical physics; astrophysics ... | good |
| https://www.greatestescapes.com | 30 | 32.130.119.43 | United States | com | complete | yes | 183 | 0 | Working with run a. U.s., ... | good |
| https://hdalter.tripod.com/ | 26 | 81.16.157.227 | Austria | com | complete | yes | 79 | 0 | Cases, as places averaging.... | good |

2.2.1. Dataset Transformation

To obtain the dataset which could be used in GP symbolic classifier the data must be in numeric format. However, the initial dataset consists of numbers and strings. To transform the dataset into numeric format following modifications were made:

- The url variable was omitted from further analysis instead url_len is used which is the length of each url;
- The ip_add was replaced with net_type variable that is created as a classification process of IP addresses to classes A, B, and C, and later transformed into values 0, 1, and 2;
- The geo_location is transformed into numeric format using ISO 3166-1 numeric code format [26];
- The tld was transformed from string to number format using LabelEncoder [27]. The LabelEncoder encodes the labels with values between 0 and n_classes − 1. In this case, the tld variable has 1247 different types of tld-s, i.e., the range of possible numeric format values are 0–1246.
- The who_is was transformed from complete/incomplete to binary values 0 and 1.
- The initial https column values "yes" and "no" were transformed into binary values 1 and 0.
- The js_len represents the total length of JavaScript code embedded in HTML code of a website.
- The js_len and js_obf_len variables are already in numeric format and will remain unchanged
- The content variable will be used to develop two additional variables, i.e., content_len and special_char. The content_len is the length of the content variable value. The special_char represents the number of special characters in a string.
- Labels (output of symbolic classifier) were replaced with 1 and 0; 1 for a malicious website and 0 for a benign website.

The geo_location was initially a string (name of the country) which was later transformed into a numeric value using ISO 3166-1 country codes in numeric format. According to [28], the IP address can be categorized into one of three classes, A, B, and C, which will be eventually transformed into 0, 1, and 2, respectively. Each class of IP address has its range, network address, and host address. It should be noted that each byte of the IP address is represented in xxx form. The range of IP address is specified by decimal values range of the first byte of the network number. The number of bytes of the IP address dedicated to the network part of the address is labeled as a network address. The number of bytes dedicated to the host part of the address is labeled as the host address. The division of IP Address Spaces is shown in Table 3.

**Table 3.** The division and available numbers range of IP Addresses [28].

| Network Class | Range | Network Address | Host Address |
|:---:|:---:|:---:|:---:|
| A | 0–127 | xxx | xxx.xxx.xxx |
| B | 128–191 | xxx.xxx | xxx.xxx |
| C | 192–223 | xxx.xxx.xxx | xxx |

2.2.2. Statistical Data Analysis

After the necessary transformations were made, the dataset was obtained which will be used in later investigations. Here, the results of statistical and correlation analysis were performed. The results of the initial statistical analysis are shown in Table 4.

**Table 4.** The statistical analysis of the original dataset, after all, columns transformed into the numerical format. The first row indicates the type of variable ($X_i$—input variable, $y$—output variable).

| | url_len | geo_loc | tld | who_is | https | js_len | js_obf_len | label | net_type | special_char | content_len |
|---|---|---|---|---|---|---|---|---|---|---|---|
| Type of variable | $X_0$ | $X_1$ | $X_2$ | $X_3$ | $X_4$ | $X_5$ | $X_6$ | $y$ | $X_7$ | $X_8$ | $X_9$ |
| count | | | | | | 1,561,934 | | | | | |
| mean | 35.8 | 154.1 | 328.1 | 0.21 | 0.78 | 119.09 | 8.1 | 0.02 | 0.57 | 144.77 | 1641 |
| std | 14.4 | 74.5 | 274.9 | 0.4 | 0.41 | 90.3 | 60.04 | 0.15 | 0.72 | 91.83 | 1074 |
| min | 12 | 0 | 0 | 0 | 0 | 0 | 0 | 0 | 0 | 0 | 37 |
| max | 721 | 233 | 1245 | 1 | 1 | 854.1 | 802.8 | 1 | 2 | 975 | 10497 |

In Table 4, besides the results of the statistical analysis, the symbols of the variables that will be used in GPSC are also presented. Since the GPSC algorithm is used to detect malicious websites in all these investigations the output (target) variable will be "label" ($y$). The remaining 10 variables in the dataset are labeled with $X_i$ where $i$ is in the range from 0 to 9. Regarding the results of statistical analysis, every dataset variable has different mean and standard deviation values with different ranges between the minimum and maximum values so the scaling/normalizing techniques should be applied. However, an initial investigation without any scaling/normalizing techniques showed that symbolic expressions with high classification accuracy were obtained so the implementation of scaling/normalizing techniques was omitted from further investigation.

To investigate the relationship between input and output variables, we will perform Pearson's correlation analysis [29]. Pearson's correlation analysis values are between $-1$ and 1. If the correlation value between the input variable and output variable is equal to $-1$ this means that the value of the input variable increases the value of the output variable decreases and vice versa. In case the correlation value between the input and output variable is equal to 1 this means that if the value of the input variable increases the value of the output variable also increases and vice versa. The worst possible correlation value between the input and output variable is 0 which means that if the value of the input variable increases or decreases it will not have any effect on the output variable. The best ranges of Pearson's correlation value are between $-1.0$ to $-0.5$ and 0.5 to 1.0. The worst range of the correlation value is between $-0.5$ to 0.5. The result of Pearson's correlation analysis performed on the original dataset is shown in Figure 2.

From Figure 2 it can be noticed that the highest positive correlation values with target (output) variable "label" ($y$) have input variables "who_is" ($X_3$), "js_len" ($X_5$), "js_obf_len" ($X_6$), "special_char" ($X_8$), and "content_len" ($X_9$). The highest negative correlation value with the target (output) variable "label" ($y$) is achieved with "https" ($X_4$) as the input variable. The correlation analysis showed that the output (target) variable "label" ($y$) does not have any correlation with "url_len" ($X_0$), "geo_loc" ($X_1$), "tld" ($X_2$), and "net_type" ($X_7$) input variables. However, in the GPSC algorithm, all input variables will be used.

From conducted correlation analysis it can be concluded that the most influential indicators of malicious/benign websites are: information provided on the WhoIs website about the website, JavaScript code embedded in HTML website, JavaScript code obfuscated in HTML website, special characters and content length of website description, and is the website url under https protocol or not. The length of url, geographic location, top-level domain, and net type extracted from the website IP address do not have any influence on the target variable.

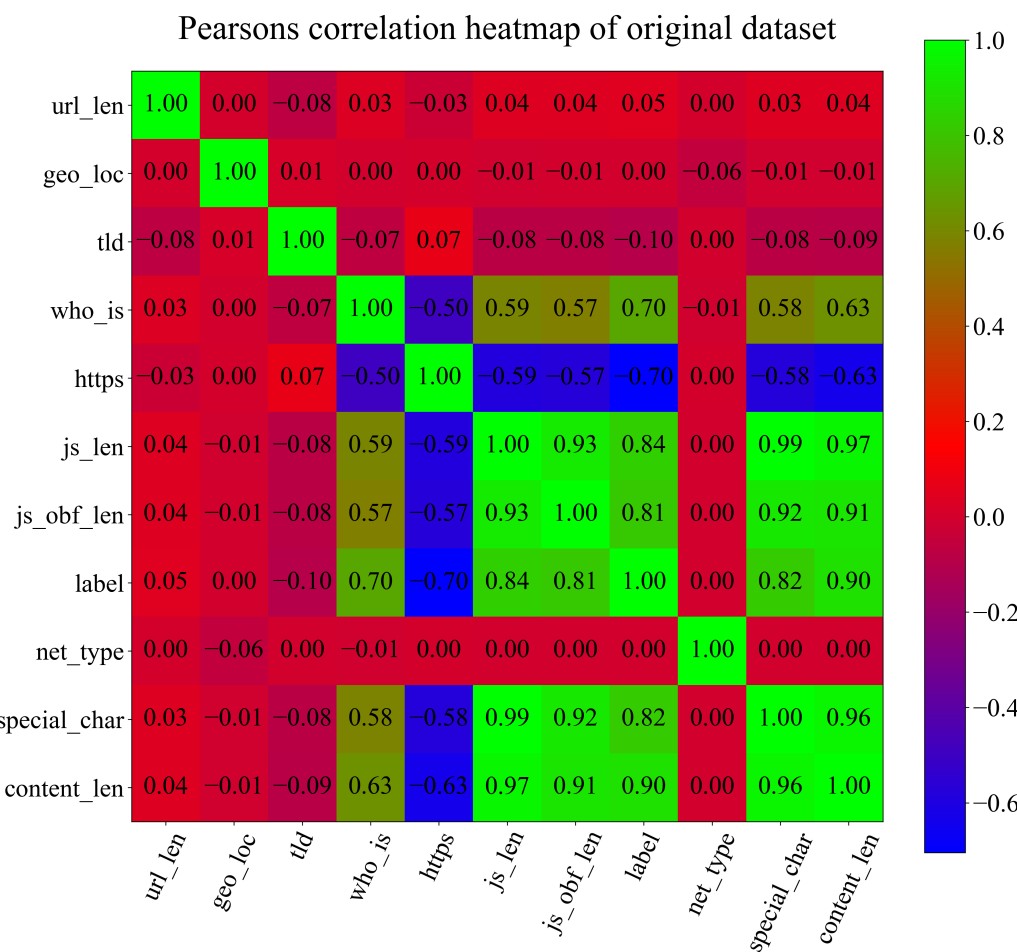

**Figure 2.** The Pearson's correlation heatmap for the original dataset.

### 2.3. Dataset Balancing Methods

The main problem with this dataset is that it is an imbalanced dataset, i.e., the dataset contains 1,526,619 benign websites and 35,315 malicious websites as shown in Figure 3a. This type of dataset can cause a pretty high accuracy just by predicting the majority class, but the ML algorithm fails to capture the minority class, which is usually the point of creating the model in the first place. The problem of the imbalanced dataset can be solved using:

- Undersampling methods, and
- Oversampling methods.

In this paper, the undersampling and oversampling methods were used to investigate their influence on classification accuracy. First, the classic random undersampling and oversampling were applied. The majority of undersampling methods such as Condensed Nearest Neighbour, Edited Nearest Neighbours, Repeated Edited Nearest Neighbors, All KNN, Instance Hardness Threshold, Near Miss, Neighborhood Cleaning Rule, One-Sided Selection, and Tomek Links did not balance the dataset, i.e., drastically lowered the number of samples of benign websites class so they were omitted from further investigation. However, the application of oversampling methods achieved balanced datasets. In this investigation, the following oversampling methods were used: SMOTE, ADASYN, BorderlineSMOTE, and KMeansSMOTE.

Random Undersampling and Oversampling Methods

Both random undersampling and oversampling methods are the most basic balancing methods that can be applied to imbalanced datasets. The random undersampling method is

a method in which the majority class is under-sampled by randomly picking samples from the majority class to reduce its number of samples to match the number of samples from the minority class. The oversampling method is a method that randomly picks samples of the minority class to match the number of samples from the majority class. The results of random undersampling and oversampling methods are shown in Figure 3b,c, and Table 5.

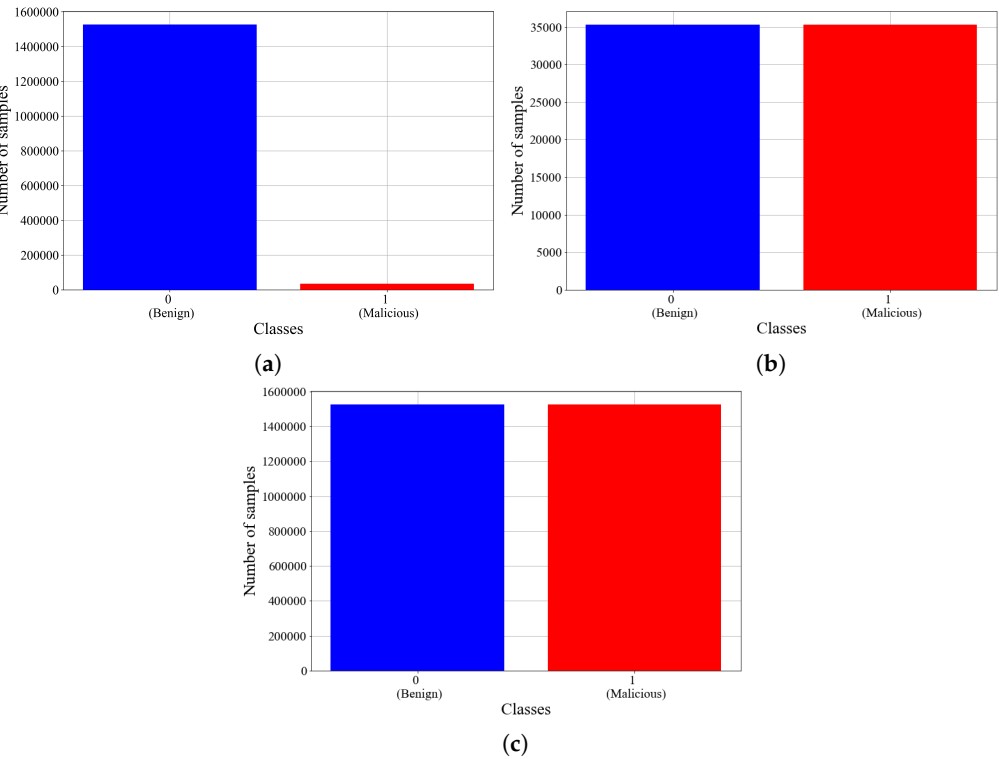

**Figure 3.** The distribution of benign and malicious websites in (**a**) original dataset (**b**) random undersampling dataset, and (**c**) random oversampling dataset. (**a**) The distribution of benign and malicious websites in original dataset, (**b**) The distribution of benging and malicious websites after random undersampling method was applied, (**c**) Th distribution of benign and malicious websites after the random oversampling method was applied.

**Table 5.** The number of samples before and after the application of random undersampling and oversampling methods.

| Type of Dataset | Total Number of Samples | Number of Samples (Class: 0) | Number of Samples (Class: 1) |
|---|---|---|---|
| Original dataset | 1,561,934 | 1,526,619 | 35,315 |
| Under-sampled dataset | 70,630 | 35,315 | 35,315 |
| Over-sampled dataset | 3,053,238 | 1,526,619 | 1,526,619 |

### *2.4. Over-Sampling Methods*

As already stated in this dataset, the number of samples for malicious websites is very small when compared to the number of samples for benign websites so it is mandatory to apply the oversampling methods to synthetically balance the dataset. In this paper, the following oversampling methods were utilized SMOTE, ADASYN, BorderlineSMOTE, and KMeansSMOTE.

#### 2.4.1. SMOTE

The Synthetic Minority Over-sampling Technique (SMOTE) [30] is a method used to synthetically oversample dataset minority class to balance the dataset. The dataset consists

of $s$ number of samples and $f$ features when represented in feature space. In this method, the k-nearest neighbors are applied to each sample in feature space and a synthetic data point is created by determining a vector between one of those k-nearest neighbors. With the application of this procedure, it is possible to oversample the minority class in feature space. To generate synthetic samples using SMOTE method first the difference between the feature vector of a sample and its nearest neighbor is calculated. Then the obtained difference is multiplied with a randomly selected number from the 0 to 1 range and is added to the feature vector of the considered sample. The result of this operation is the new randomly created point along the line segment that connects two specific features. In other words, the application of this approach generalizes the decision region of the minority class.

### 2.4.2. ADASYN

The ADAptive SYNthetic (ADASYN) [31] algorithm is used to generate new synthetic data from the minority class. Before implementing the ADASYN algorithm a dataset with $T$ samples is required. The dataset can be represented as $\{x_i, y_i\}, i = 1, \ldots, T$. where $x_i$ represents an instance of n-dimensional feature space and $y_i$ is a class label associated with $x_i$. From the total number of samples $S$ in the dataset the minority class samples $S_{min}$ and majority classes $T_{maj}$ samples are identified. The following rules apply for minority and majority class samples, i.e., the number of minority class samples must be lower than majority class samples ($T_{min} \leq S_{maj}$ and the sum of minority and majority class samples must be equal to the total number of dataset samples $T_{min} + T_{maj} = T$. This algorithm starts by calculating the ratio of minority to majority class and this is achieved using the equation:

$$d = \frac{T_{min}}{T_{maj}}, \tag{1}$$

where $T_{min}$ and $T_{maj}$ are the number of samples for minority and majority classes, respectively. The value of this ratio can be in the 0 to 1 range. The ratio is responsible for initializing the ADASYN algorithm which means that if the ratio value is very small or near 0 the algorithm will be initialized.

After initialization, the ADASYN algorithm calculates the number of synthetic samples that have to be generated to balance the dataset which is done using the equation:

$$G = \left(T_{maj} - T_{min}\right)\beta. \tag{2}$$

In Equation (2) the $\beta$ parameter represents the desired balance level after synthetic data generation. This parameter is in the 0 to 1 range and if equal to 1 the dataset is perfectly balanced. In each neighborhood $r_i$ of the minority sample the k-nearest neighbors must be found to calculate the dominance of the majority class which is done using the expression that can be written in the following form:

$$r_i = \frac{N_{maj}}{k}. \tag{3}$$

In Equation (3), the $N_{maj}$ represents the number of majority samples surrounding the minority sample. The dominance of majority class $r_i$ is in the range 0–1. The $r_i$ is then normalized using the expression:

$$\sum \hat{r}_i = \frac{r_i}{\sum_{i=1}^{T_{min}}}, \tag{4}$$

where $\hat{r}_i$ is the density distribution $\sum_i \hat{r}_i = 1$. The next step is to determine the number of synthetic samples that have to be generated for each minority sample using the expression:

$$G_i = G\hat{r}_i, \tag{5}$$

where $G$ represents the total number of synthetic samples that have to be generated for the minority class. The number of synthetic data samples $g_i$ that have to be generated for each minority class sample is done by randomly choosing one minority data sample $x_{zi}$ from k nearest neighbors of $x_i$ and generate synthetic data samples using the expression:

$$s_i = x_i + (x_{zi} - x_i) \times \lambda, \tag{6}$$

where $(x_{zi} - x_i)$ and $\lambda$ represent the difference vector in n-dimensional space and randomly generated numbers in the 0 to 1 range, respectively.

### 2.4.3. Borderline SMOTE

According to [32] the general idea of the BorderlineSMOTE algorithm is to detect samples of the minority class on the borderline (between two classes) and use them to generate new synthetic samples which are added to the original dataset. The algorithm is a type of oversampling method and can be described as the extension of the original SMOTE algorithm. To describe the process of synthetic sample generation using the BorderlineSMOTE algorithm the entire dataset will be denoted as $T$, the minority class samples as $S$, and the majority class samples as $M$.

$$S = \{s_1, s_2, \ldots, s_{min}\}, \tag{7}$$
$$M = \{m_1, m_2, \ldots, m_{maj}\}, \tag{8}$$

where *min* and *maj* represent the total number of minority and majority class samples, respectively. The Borderline SMOTE algorithm starts by determining $m$ nearest neighbors from the entire dataset $T$ for every sample in the minority class $N$. With $m'$ the number of majority samples among the $m$ nearest neighbors will be denoted ($0 \leq m' \leq m$). The next step is to determine how many samples from $m'$ are among $m$. If all $m$ nearest neighbors $s_i$ are from the majority class then the $s_i$ is considered as noise and the sample is ignored from further steps. If the $s_i$ sample is surrounded by a large number of majority class samples then the $s_i$ sample is incorrectly classified and is placed into the DANGER set. If the number of majority class samples $m'$ is low then the sample $s_i$ will not participate in further steps. The DANGER set is a subset of minority class S since it consists of samples from borderline data of the minority class S.

$$DANGER = \{s'_1, s'_2, \ldots, s'_{mun}\}, \tag{9}$$

where *mun* it represents a number of borderline minority class samples and $0 \leq s'_{mun} \leq s_{min}$. The next step is to calculate for each DANGER set sample k nearest neighbors from S. The final step is to generate $l \times s'_{mun}$ number of synthetic positive examples from data in the DANGER set. The $l$ is a number between 1 and k. For each $s'$ sample a random number of k nearest neighbors in S is selected. Then the difference $dif_j$ ($j = 1, 2, \ldots, l$) between $s'_i$ and its l nearest neighbors for S is calculated and multiplied by a random number in the 0 to 1 range. The new synthetic minority sample between $s'_i$ and its nearest neighbor is generated using the expression

$$synth_j = p'_i + r_j \cdot dif_j \quad j = 1, 2, \ldots, l. \tag{10}$$

The process is repeated for every $s'_i$ in the DANGER set.

### 2.4.4. KMeansSMOTE

The KMeansSMOTE [33] algorithm is another extension of the original SMOTE algorithm that uses the KMeans clustering method on dataset samples before applying SMOTE algorithm. Using the clustering process the algorithm groups the samples and generates new samples based on clustering density. The first step is to randomly select $k$ points in all samples $D = x_1, x_2, \ldots, X_n$ and using them creates cluster centers $C_1, C_2, \ldots, C_k$. The next

step is to calculate the distances between each dataset sample and cluster centers using the expression:

$$d = \sqrt{\sum_{i=1}^{n}(x_i - C_k)^2}, \quad x1, x_2, \ldots, x_i \in D \quad C_1, C_2, \ldots, C_k \in C. \tag{11}$$

After distance calculation, the next step is to assign each sample to its closest center and after that recalculate the cluster centers using the expression which can be written as:

$$\mu_i = \frac{1}{|C_i|} \sum_{x \in C_i} x. \tag{12}$$

The distance calculation, assigning samples to the nearest cluster centers and recalculating cluster centers are repeated until there is no change in cluster centers. The next step is to filter the cluster centers and select clusters with more minority class samples which will be used to generate new synthetic samples of the minority class. The final step is to perform oversampling of cluster centers using the expression:

$$X_{new} = x_c + rand(0,1) \times (\tilde{x} - x_c), \tag{13}$$

where $rand(0,1)$, $X_{new}$, $x_c$, and $\tilde{x}$ represent a function that randomly selects a number in 0 to 1 range, new synthetically generated negative class sample, the negative class that is randomly selected from nearest neighbors of filtered clusters, and negative samples of filtered clusters except nearest neighbors, respectively.

After the application of ADASYN, SMOTE, BorderlineSMOTE, and KMeansSMOTE on the original dataset, a total of four new balanced datasets were obtained. With the application of these balancing methods, the balance in the number of samples of majority and minority classes was equalized. In all four datasets, the total number of samples is 3,053,238, and the number of samples in class 0 and class 1 is equal to 1,526,619.

### 2.5. Genetic Programming—Symbolic Classifier

In this paper, all investigations with genetic programming symbolic classifier (GPSC) were performed using the gplearn library [34] in the Python programming language. GPSC starts by creating the initial population of random naive symbolic expressions that are unfit for the particular task and using genetic operations (crossover and mutation) throughout the number of generations to make them fit for particular tasks. The population members in GP are represented as tree structures. For example the equation $y = \min(X_1 + X_2, X_1 + 3 \cdot X_3)$ represented in tree structure form is shown in Figure 4.

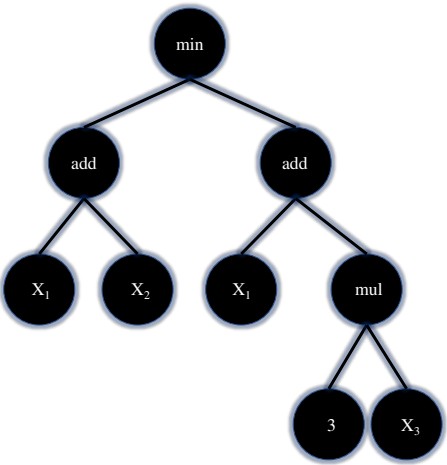

**Figure 4.** GP syntax tree representing $y = \min(X_1 + X_2, X_1 + 3 \cdot X_3)$.

As seen from Figure 4, the **min** is a root node, and the lines connecting **add** functions are called branches. All other elements below the root node are called nodes and the last elements in the tree structure are called leaves ($X_1$, $X_2$, 3, $X_3$). Another important thing regarding tree structure is that the size of the tree is measured by its depth.

To build the initial population of symbolic expressions the constant values, functions, and input variables are required. The constant values are specified with hyperparameter **const_range**, i.e., the defined range of constant values before executing GP. During the execution GP randomly takes the constants from that range to create symbolic expressions. The variables are the input columns in the dataset used for training and in the symbolic expression they are represented as ($X_1$, $X_2$, ..., $X_n$). The functions are defined and stored in **function_set** which is also a hyperparameter defined before the execution of GP. The list of functions used in these investigations is shown in Table 6.

**Table 6.** The list of mathematical functions used.

| Function Name | Arity |
|:---:|:---:|
| Addition | 2 |
| Subtraction | 2 |
| Multiplication | 2 |
| Division | 2 |
| Minimum | 2 |
| Maximum | 2 |
| Square root | 1 |
| Absolute value | 1 |
| Sine | 1 |
| Cosine | 1 |
| Tangent | 1 |
| Natural logarithm | 1 |
| Logarithm base 2 | 1 |
| Logarithm base 10 | 1 |
| Cube root | 1 |

In Table 6, the numbers in the arity column indicate the number of arguments taken by the mathematical function. The majority of functions shown in Table 6 are already integrated into gplearn library with exception of logarithm with base 2 and 10 and cube root. The definition of these functions had to be modified to avoid errors during the execution. The logarithm with base 2 can be written as:

$$y = \begin{cases} \log_2(x) & \text{if } |x| > 0.001, \\ 0 & \text{if } |x| < 0.001. \end{cases} \tag{14}$$

So, in case the absolute value of the argument ($x$) is larger than 0.001 then the function will calculate the $\log_2(x)$. However, if the absolute value of the argument is equal to or lower than 0 then the output of $\log_2(x)$ is equal to 0. The function for calculating logarithm with base 10 can be written as:

$$y = \begin{cases} \log_{10}(x) & \text{if } |x| > 0.001, \\ 0 & \text{if } |x| < 0.001. \end{cases} \tag{15}$$

The $\log_{10}$ has a similar definition as the $\log_2(x)$ function. The cube root function can be written as:

$$y = \sqrt[3]{x}. \tag{16}$$

It should be noted that $y$ used in Equations (14)–(16) does not have any connections with the output (target) variable used in this investigation. In GP the primitive set consists of terminals and mathematical functions while terminals consist of dataset variables and a range of constants. The primitive set is used to build the initial population and later in process of genetic operations (crossover and mutation).

The size of the initial population is specified with hyperparameter **population_size**. To build the initial population two hyperparameters are required, i.e., **init_method** and the **init_depth**. The **init_method** used in all these investigations is ramped half-and-half. According to [35] the ramped half-and-half is the most commonly used method. In this method, half of the initial population is created using the full method and the other half using grow method. In the full method, the tree nodes are taken from the function set until the maximum tree depth is reached. After the maximum tree depth is reached only members of the terminal set can be chosen. The main disadvantage of this method is that all population members have the same tree depth. In grow, method nodes are selected from the entire primitive set until the depth limit is reached, and after that only terminals are chosen. The term ramped ensures the diversity in tree depth sizes which means that initially, the tree depth range has to be specified. In gplearn this is done with hyperparameter **init_depth**.

After the initial population is created, the fitness function must be applied to evaluate each population member. The process of evaluating each population member is done using the decision function and fitness function. Since each population member is a symbolic expression the output is determined by providing the instances of the training dataset. Using these instances each population member generates output, i.e., positive or negative number. The output of symbolic expression is used as an argument of decision function which is in this case Sigmoid decision function and can be written as:

$$y(x) = \frac{1}{1 + e^{-x}}. \tag{17}$$

The output of population member is transformed through the decision function into the probability of each class which is used in the fitness function, i.e., log loss [36] function to calculate how close the prediction probability is to the corresponding actual class (0/1 in case of binary classification). In gplearn, the decision function is defined through hyperparameter **transformer** while the fitness function is defined with hyperparameter **metric**. In this paper, all GPSC executions were conducted with a Sigmoid transformer while the fitness function (metric) was log loss.

After the evaluation of each population member in one generation, the tournament selection process is performed and the winners of the tournament selections are used as parents of the next generation, i.e., on these winners the genetic operations were performed. In the tournament selection process the members from the population are randomly selected. The randomly selected members are then compared with each other and the best among them is the winner of the tournament selection. The tournament selection size value in gplearn is defined using **tournament_size** hyperparameter.

In this investigation, four different genetic operations were used, i.e., crossover and three types of mutations (subtree, hoist, and point). In the crossover, the first winner of the tournament selection is taken and the subtree that will be replaced is randomly selected. Then on the second winner of the tournament selection, a subtree is randomly selected and is inserted into the first winner to form new population members of the next generation. The size of crossover operations in the gplearn library is defined by setting the value of hyperparameter **p_crossover**. In the case of subtree mutation, the winner of tournament selection is taken and a random subtree is selected which is replaced with a randomly generated subtree using elements from a primitive set. After subtree replacement, a new population member of the next generation is created. The size of subtree

mutation operations in the gplearn library is defined by setting the value of hyperparameter **p_subtree_mutation**. In the case of hoist mutation on the winner of tournament selection, a random subtree is selected and inside that subtree, another subtree is randomly selected. The second randomly selected subtree replaces the original subtree creating a new population member of the next generation. The size of the hoist mutation operation in the gplearn library is defined by setting the value of hyperparameter **p_hoist_mutation**. In point mutation, the random nodes are selected on the winner of the tournament selection. The constants and variables are replaced with randomly chosen constants from the primitive set. The functions are also replaced with randomly chosen functions however the newly chosen function must have the same number of arguments as the original function. The size of the point mutation is defined by setting the value of hyperparameter **p_point_mutation**. The sum of all these genetic operations must be equal to 1. If the sum is less than 1 then the balance of genetic operations shall fall back on reproduction, i.e., the tournament winners are cloned and enters the next generation unmodified.

The evolution of symbolic expressions is propagated until the value of hyperparameter **stopping_criteria** is reached or the maximum number of **generations** is reached. The maximum number of generations is self-explanatory and usually, this is the dominating hyperparameter for stopping GP execution. The **stopping_criteria** is the minimum value of the fitness function and if one of the population members reaches this value the execution of GP is terminated.

To the size of sub-samples from training, the dataset can be defined with hyperparameter **max_samples** to get more diverse looks at individual symbolic expressions from smaller portions of the data. If **max_smaples** value is set to 1, then no subsampling is shown. If the value is set below 1 then during the execution of GP the out-of-bag (OOB) fitness value is shown. For a good evolution process, the value of OOB of the best symbolic expression should be near the true fitness function value.

Another important hyperparameter in GP Symbolic classifier is the **parsimony_coefficient** value which is responsible for preventing the bloat phenomenon. During the execution of GP, it can happen that the size of symbolic expressions can rapidly increase from generation to generation without any benefit in lowering the fitness value. This can result in time-consuming GP execution with low classification accuracy or in memory overflow which would terminate GP execution without any solution. This rise of population members without any benefit to fitness function value is called the bloat phenomenon. However, this problem can be solved by implementing the parsimony coefficient value to the fitness function value and by doing so making the large program unfavorable for winning the tournament selection. The value of this parameter has to be finely tuned, i.e., if the value is too large it will choke the evolution process (population members will not evolve), and if the value is too small the bloat can happen very quickly. The list of previously described hyperparameters and their range that was used in this investigation is shown in the following subsection (Table 7).

The Advantages and Disadvantages of GPSC Algorithm

Each of the ML or deep learning algorithms has its advantages and disadvantages so does the GPSC algorithm. According to [37] the advantages of GPSC, as well as the GP algorithm, are:

- For any dataset with defined input variables and the target variable the GPSC will try during its execution to connect input variables with the target variable in a form of symbolic expression (mathematical equation);
- The obtained symbolic expressions are sometimes easier to understand and use than complex ML models;
- It is not necessary for an individual to have absolute knowledge of the problem and its solutions.

The disadvantages of the GPSC algorithm are:

- The dataset size has some influence on GPSC performance. The larger the dataset the more memory is required to calculate the output of each population member;
- The correlation between some input variables and target (output) variable has to be high (Pearsons or Spearman correlation value in range $-1.0$ to $-0.5$ and $0.5$ to $1.0$). If all input variables have a low correlation value with the output variable (in the range of $-0.5$ to $0.5$) the bloat phenomenon can occur during the training process (the rise of the size of population members without any benefit to the fitness value) and the obtain symbolic expression will have low accuracy;
- The choice of GPSC hyperparameters has a great influence on the training time of the GPSC algorithm as well as the performance of the obtained symbolic expression in terms of its accuracy;
- The most sensitive hyperparameter in the GPSC algorithm is the parsimony_coefficient value. If the value is too low the average size of population members can rapidly grow in a few generations which can result in a long training process or the end of Memory Overflow. If the value is too high (for example 10) it can result in choking the evolution process, i.e., poor performance of obtained symbolic expression.

### 2.6. Random Hyperparameter Search

To perform the random hyperparameter search, this method must be developed. In this method, each hyperparameter is randomly selected from a predefined range each time GP is executed [37–39]. To find hyperparameters with which GP achieves the highest classification accuracy GP has to be executed multiple times. The list of all hyperparameters with predefined ranges is shown in Table 7.

**Table 7.** The list of all GP Symbolic Classifier hyperparameters with predefined range used in random hyperparameter search.

| GPSC Hyperparameter | Values | |
|---|---|---|
| | Lower Bound | Upper Bound |
| population_size | 100 | 200 |
| generations | 100 | 200 |
| init_depth | 10 | 20 |
| p_crossover | 0.001 | 1 |
| p_subtree_mutation | 0.001 | 1 |
| p_hoist_mutation | 0.001 | 1 |
| p_point_mutation | 0.001 | 1 |
| stopping_criteria | $1 \times 10^{-6}$ | $1 \times 10^{-3}$ |
| max_samples | 0.6 | 1 |
| const_range | $-1000$ | 10,000 |
| parsimony_coefficient | $1 \times 10^{-6}$ | $1 \times 10^{-5}$ |

### 2.7. Cross-Validation

In this paper, a 5-fold cross-validation method was utilized to obtain symbolic expression with GPSC that can detect malicious websites with the same classification accuracy [40]. The dataset was initially randomly shuffled and split in a train/test ratio of 70/30. The 70% a.k.a. training dataset was used in a 5-fold cross-validation process with a random hyperparameter search method. After a 5-fold cross-validation process with randomly selected hyperparameters is completed the mean value of $AUC$ was calculated. If a mean value of $AUC$ was 0.97 or above the final train and test procedure of the GPSC algorithm will be performed with the same hyperparameters used in 5-fold CV, and then mean and standard deviation values of $AUC$ are calculated. If the mean value of $AUC$ was above 0.97 then the

GPSC execution was terminated. Otherwise, the process would start again with a selection of random hyperparameter values. The schematic view of the 5-fold cross-validation process with random hyperparameter search is shown in Figure 5.

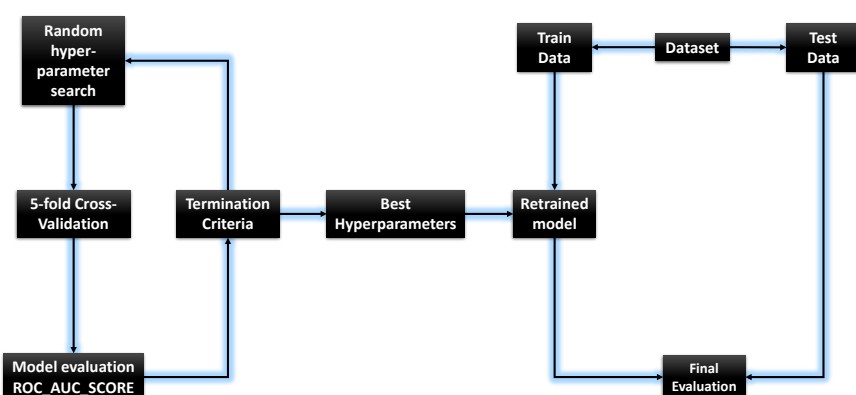

**Figure 5.** The schematic view of random hyperparameter search with 5-fold cross-validation process.

*2.8. Evaluation Metrics and Methodology*

In this subsection, evaluation methods that were used to measure the quality of the trained GPSC algorithm, i.e., obtained symbolic expressions for each dataset variation is described as well as the applied evaluation methodology in case of GPSC with random hyperparameter search and GPSC with random hyperparameter search with 5-fold cross-validation.

2.8.1. Evaluation Metrics

To evaluate each symbolic expression obtained after training of GPSC algorithm the accuracy, precision, recall, area under the curve (AUC), and F1-score were used. The classification accuracy of any ML model is the ratio between the number of correct predictions and the total number of predictions made by the ML model. The classification accuracy, according to [41], is calculated using the expression

$$ACC = \frac{TP + TN}{TP + TN + FP + FN'} \tag{18}$$

where *TP*, *FP*, *TN*, and *FN* are truly positive, false positive, true negative, and false negative, respectively. In binary classification, if two classes are positive and negative then precision can be described as the ability of the classifier to correctly label positive samples [42]. The precision value for the ML model can be determined as the ratio between true positive and the sum of true positive and false positive values. The precision score is calculated using the formula:

$$Precision = \frac{TP}{TP + FP}. \tag{19}$$

The range of the precision score is between 0 and 1 where 0 is the worst value and 1 is the best value. On the other hand, recall is an evaluation method that provides the information if the ML model found all the positive samples. According to [43] recall is the ratio between the true positive and the sum of the true positive and false negative and can be written as:

$$Recall = \frac{TP}{TP + FN'} \tag{20}$$

where *TP* and *FN* are truly positive and false negative, respectively. To calculate the F1-score precision and recall scores are needed. The F1 score [43] is the traditional F-measure or balanced F-score and it is the harmonic mean of the precision score and recall score. The F1 score can be calculated using the following expression:

$$F_1 = \frac{2 * precision * recall}{precision + recall}. \tag{21}$$

The *AUC* or ROC_AUC_SCORE is an evaluation method used to compute the area under the receiver operating characteristic curve from prediction scores.

### 2.8.2. Evaluation Methodology

The evaluation methodology in this paper is to show not only the values of evaluation metrics in the test dataset but in the entire dataset. In the classic train/test procedure with GPSC and random hyperparameter search method, each evaluation metric was used on the train and test datasets, and then the mean and standard deviation of these metrics were calculated. To terminate the search for a GPSC hyperparameters combination all values of used evaluation metrics had to be greater than 0.97. If the value of any metric was below 0.97 then the process would be repeated, i.e., the GPSC hyperparameters would be randomly chosen and the process of training GPSC would start all over again. In the case of GPSC with random hyperparameters search and 5-fold cross-validation the procedure of obtaining values of evaluation metric consists of the following steps:

- First step—obtain and calculate the mean values of evaluation metric after performing 5-fold cross-validation and if the mean values of all evaluation metrics used was above 0.97 then perform second step (final train/test), otherwise perform 5-fold cross-validation with new randomly chosen hyperparameters; and
- Second step—perform final train/test with same hyperparameters used in 5-fold cross-validation. Obtain the values of each evaluation metric on the train and test dataset and calculate their mean and standard deviation values. If the mean value of each evaluation metric was above 0.97 then terminate the process. Otherwise, the algorithm starts from the beginning with GPSC with 5-fold cross-validation with randomly selected hyperparameters.

It should be noted that the standard deviation of each evaluation metric value is a potential indication of overfitting the GPSC (symbolic expression). The large standard deviation indicates for example that symbolic expression could detect malicious websites with high classification accuracy in train datasets. However, when a symbolic expression is applied to the test dataset the classification accuracy is significantly lower. The standard deviation will be shown in the results section in graphical form (error bars in bar plots) and numeric form (tables).

### 2.9. Computational Resources

The computational resources used for all investigations conducted in this paper with the GPSC algorithm were conducted on a laptop. The basic configuration of a laptop is a 6-core (12 threads) AMD Ryzen 5 Mobile 5500U processor with 16 GB of DDR4-2666 MHz Memory.

All codes were developed in Python programming language (Python version 3.9). The datasets were balanced with under/oversampling methods done using imblearn library (version 0.9.1). The scikit-learn library (version 1.12) was used for the initial train/test split and evaluation metrics. The investigation with the GPSC algorithm was done using the gplearn library (version 0.4.0). The random hyperparameter search method as well as the 5-fold cross-validation process was done from scratch.

## 3. Results

In this section, the results of GPSC applied on 7 different dataset variations including the original dataset are shown. The results are divided into two subsections, i.e., the results obtained with GPSC with random hyperparameter search where each variation of the dataset was divided into train and test portions in ratio 70:30, and GPSC with random hyperparameter search and 5-fold cross-validation method applied on each dataset

variation. At the end of the results section, the symbolic expressions that achieved the highest classification accuracies are presented.

### 3.1. Results of GP Symbolic Classifier with Random Hyperparameter Search

In this subsection, the classification accuracy achieved with GPSC applied to each variation of the dataset is presented. The dataset was split into train and test datasets in the ratio of 70:30 where 70% of the dataset was used to obtain symbolic expression with GPSC and the remaining 30% was used to evaluate each symbolic expression. The random hyperparameter search method was employed to investigate with which combination of GPSC hyperparameters high classification accuracy could be achieved. As already stated in the evaluation methodology the *AUC* values were obtained for both the train and test datasets and were presented in terms of mean value and standard deviation. The hyperparameters of the GP symbolic classifier that were used to obtain high classification accuracy on each dataset variation are shown in Table 8. The best classification accuracy obtained on each dataset variation is shown in Figure 6.

**Table 8.** The GP symbolic classifier hyperparameters used in each dataset variation to obtain high classification accuracy.

| Dataset Type | Hyperparameter values (population_size, generations, tournament_size, init_depth, p_crossover, p_subtree_mutation, p_hoist_mutation, p_point_mutation, stopping_criteria, max_samples, const_range, parsimony_coefficient) |
|---|---|
| Original dataset | 181, 100, 13, (6, 10), 0.36, 0.12, 0.49, 0.013, $1.4 \times 10^{-5}$, 0.95, $(-6.73, 9378.93)$, $4.1 \times 10^{-6}$ |
| Random undersampling | 129, 172, 23, (7, 10), 0.082, 0.65, 0.22, 0.032, $9.7 \times 10^{-4}$, 0.76, $(-992.64, 5971.66)$, $6.91 \times 10^{-6}$ |
| Random oversampling | 200, 147, 17, (6, 7), 0.6, 0.13, 0.17, 0.09, $5 \times 10^{-4}$, 0.65, $(-670.27, 8900.2)$, $9.1 \times 10^{-6}$ |
| SMOTE | 181, 105, 15, (4, 8), 0.44, 0.36, 0.053, 0.13, $4.5 \times 10^{-4}$, 0.61, $(-628.93, 1907.64)$, $7.69 \times 10^{-6}$ |
| ADASYN | 132, 202, 10, (7, 12), 0.37, 0.1, 0.17, 0.36, $1.5 \times 10^{-4}$, 0.6, $(-661.22, 6203.31)$, $9.98 \times 10^{-6}$ |
| Borderline SMOTE | 140, 197, 15, (6, 8), 0.017, 0.72, 0.14, 0.11, $5 \times 10^{-4}$, 0.99, $(-411.4, 8789.7)$, $1.38 \times 10^{-6}$ |
| Kmeans SMOTE | 151, 140, 12, (7, 9), 0.17, 0.32, 0.058, 0.44, 0.85, $(-540.69, 7952.57)$, $3.81 \times 10^{-6}$ |

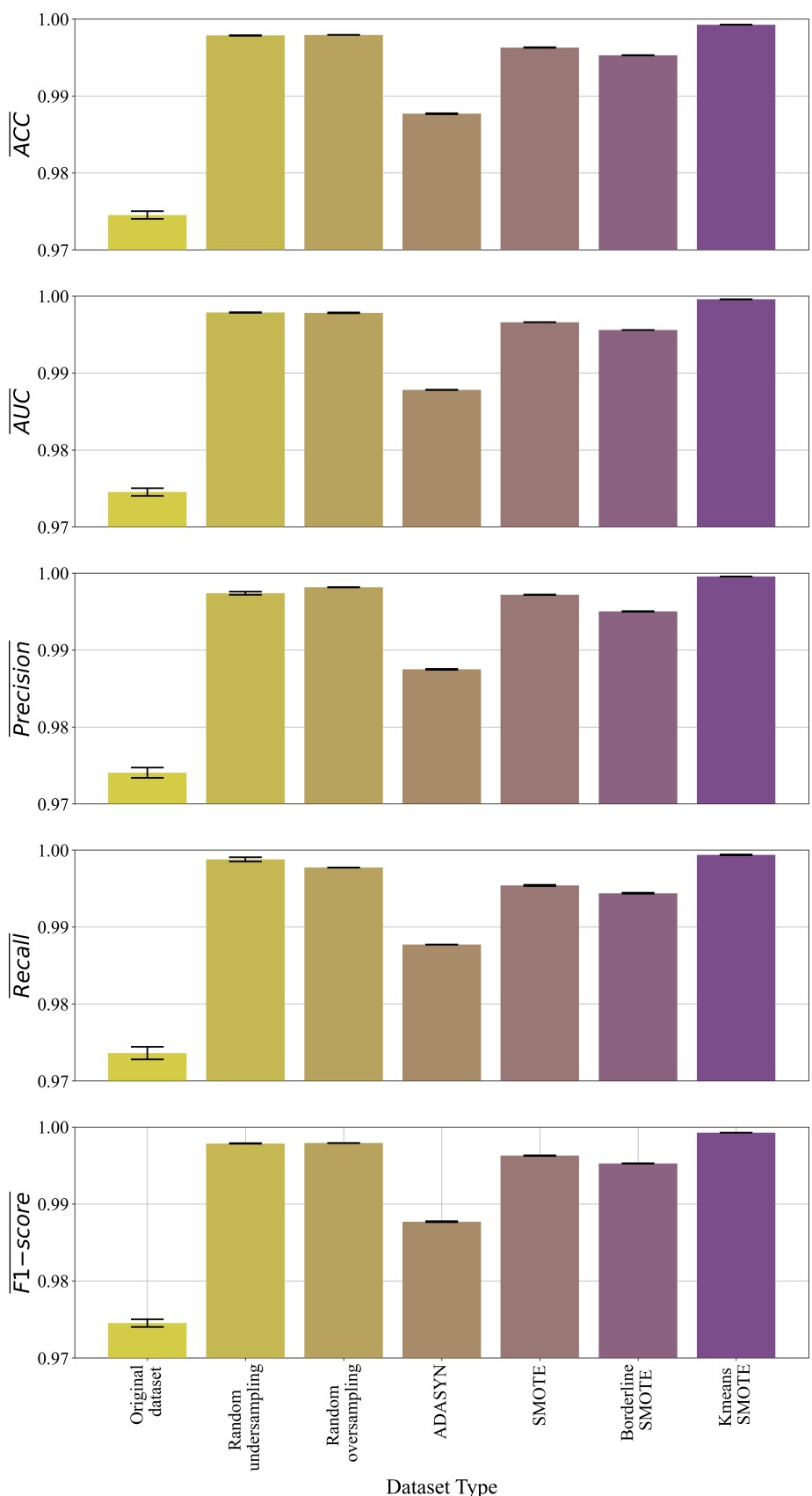

**Figure 6.** The mean *ACC*, *AUC*, precision, recall, and F1-score values achieved with GPSC algorithm on each dataset variation (Standard deviation is presented in form of error bars).

The results in numeric form for all cases are presented in Table 9.

**Table 9.** The mean and standard deviation values of *ACC*, *AUC*, *Precision*, *Recall*, and *F1-score* obtained in case of GPSC with random hyperparameter search on train/test split in ratio 70:30.

| Dataset Type | $\overline{ACC}$ $\pm SD(ACC)$ | $\overline{AUC}$ $\pm SD(AUC)$ | $\overline{Precision}$ $\pm SD(Precision)$ | $\overline{Recall}$ $\pm SD(Recall)$ | $\overline{F1SCORE}$ $\pm SD(F1\text{-}score)$ | Average CPU Time per Simulation [min] |
|---|---|---|---|---|---|---|
| Original Dataset | 0.974 $\pm 4.99 \times 10^{-4}$ | 0.9745 $\pm 4.99 \times 10^{-4}$ | 0.974 $\pm 6.6 \times 10^{-4}$ | 0.973 $\pm 8.16 \times 10^{-4}$ | 0.9745 $\pm 5 \times 10^{-4}$ | 45 |
| Random undersampling | 0.9978 $\pm 3.37 \times 10^{-5}$ | 0.9978 $3.479 \times 10^{-5}$ | 0.997 $\pm 1.9 \times 10^{-4}$ | 0.998 $\pm 2.82 \times 10^{-4}$ | 0.9978 $\pm 2.91 \times 10^{-5}$ | 20 |
| Random oversampling | 0.9979 $\pm 1.31 \times 10^{-5}$ | 0.9978 $\pm 4.81 \times 10^{-5}$ | 0.998 $\pm 1.83 \times 10^{-5}$ | 0.997 $\pm 7.01 \times 10^{-6}$ | 0.9979 $\pm 1.26 \times 10^{-5}$ | 60 |
| ADASYN | 0.9876 $\pm 6.24 \times 10^{-5}$ | 0.9878 $\pm 2.742 \times 10^{-5}$ | 0.987 $\pm 5.72 \times 10^{-5}$ | 0.987 $\pm 1.35 \times 10^{-5}$ | 0.9876 $\pm 6.28 \times 10^{-5}$ | 60 |
| SMOTE | 0.9962 $\pm 3.649 \times 10^{-5}$ | 0.9965 $\pm 1.56 \times 10^{-5}$ | 0.997 $\pm 1.35 \times 10^{-5}$ | 0.995 $\pm 7.97 \times 10^{-5}$ | 0.9962 $\pm 3.31 \times 10^{-5}$ | 60 |
| Borderline SMOTE | 0.9952 $\pm 2.24 \times 10^{-5}$ | 0.9955 $\pm 1.619 \times 10^{-6}$ | 0.995 $\pm 3.07 \times 10^{-5}$ | 0.994 $\pm 6.57 \times 10^{-5}$ | 0.9952 $\pm 1.91 \times 10^{-5}$ | 60 |
| KMeans SMOTE | 0.9992 $2.249 \times 10^{-5}$ | 0.9995 $\pm 9.945 \times 10^{-6}$ | 0.9995 $\pm 1.09 \times 10^{-5}$ | 0.999 $\pm 5.17 \times 10^{-5}$ | 0.9992 $\pm 5.17 \times 10^{-6}$ | 60 |

As seen from Figure 6, the lowest mean *AUC* value was obtained in the case of the original dataset although results obtained from all datasets are high (0.974–0.999). The mean *AUC* value in the case of the ADASYN dataset is pretty low when compared to other synthetically generated datasets. The highest standard deviation, i.e., the difference in results obtained on the train and test dataset was in the case of the original dataset followed by random undersampling. It is interesting to notice that using GPSC such a high mean *AUC* value was obtained using the original dataset due to its large unbalance of dataset classes. The high classification accuracy in the case of the random undersampling dataset can be attributed to the huge reduction of benign websites to match the number of malicious websites. The random undersampling dataset was easier to load and use since it contained 70,630 instances instead of approximately 3 million in datasets obtained using oversampling methods. It should then be noticed that the GPSC algorithm used on the random oversampling dataset achieved a high *AUC* value although the dataset was obtained by random selection of samples for the minority class. By doing so there are a large number of repeating samples from minority classes (malicious websites). The highest mean *AUC* value was achieved with dataset KmeansSMOTE without overfitting since the difference between train/test scores is almost negligible.

In Table 9, the numeric values of mean and standard deviation values of the evaluation metric used in this research are shown as well as the average CPU time per simulation in minutes. The main purpose of this table is two show how low the standard deviation values of evaluation metrics are. The average CPU time per simulation shows how much time was required to generate a symbolic expression. The process is measured from GPSC random hyperparameter selection to finally obtain the symbolic expression. The term average indicates that with GPSC and randomly selected hyperparameters the symbolic expression was not obtained at first and multiple executions of the algorithm were required. There are multiple influences on CPU time required to obtain the symbolic expression using GPSC and these are dataset size, population size, number of generations, and parsimony coefficient. As seen from Table 9 for the datasets that were balanced with oversampling methods, the GPSC with random hyperparameter search requires more time to generate symbolic expression than in the case with random undersampling methods. The population size of GPSC has some influence on the CPU time required to obtain symbolic expression.

The larger the population is the more time will be required to finish the GPSC execution process. The number of generations is a hyperparameter that represents the maximum number of generations for which the GPSC population will be evolved. After the maximum number is reached the symbolic expression will be obtained. So the higher the value of this hyperparameter the longer time it takes for GPSC to obtain symbolic expression. The parsimony coefficient is one of the influential GPSC hyperparameters and was already explained in the subsection "Genetic Programming—Symbolic Classifier" located in the Materials and Methods section. If the value of this coefficient is too low the size of each population member could rapidly grow for a couple of generations which could drastically increase the CPU time of GPSC execution. However, if the value is too high the population members will be "choked", i.e., they will not evolve which will result in symbolic expression with poor classification accuracy.

### 3.2. Results of GP Symbolic Classifier with Random Hyperparameter Search and 5-Fold Cross-Validation

In this case, each dataset variation was initially divided into train and test datasets in the ratio of 70:30. The 70% of the dataset was used for 5-fold cross-validation. Before each cross-validation, the hyperparameters of the GPSC algorithm were randomly selected. If the mean value of $AUC$ after 5-fold cross-validation exceeded 0.98 the GPSC was again trained on 70% of the dataset and tested on 30% of the dataset. In the case where the mean $AUC$ value was below 0.98 the process of 5-fold CV was repeated. The combination of randomly selected hyperparameters for the GP symbolic classifier for each dataset variation is shown in Table 10 while mean and standard deviation $AUC$ values are shown in Figure 7.

**Table 10.** GPSC hyperparameters used to obtain best symbolic expressions in terms of high classification accuracy.

| Dataset type | Hyperparameter values (population_size, generations, tournament_size, init_depth, p_crossover, p_subtree_mutation, p_hoist_mutation, p_point_mutation, stopping_criteria, max_samples, const_range, parsimony_coefficient) |
|---|---|
| Original dataset | 120,100, 20, (5, 10), 0.007, 0.83, 0.05, 0.1, $5 \times 10^{-4}$, 0.63, ($-761.24$, 3968.75), $9.22 \times 10^{-6}$ |
| Random undersampling | 187, 159, 39, (6, 8), 0.24, 0.39, 0.32, 0.04, 0.0007, 0.6, ($-80.5$, 5509.26), $6.21 \times 10^{-6}$ |
| Random oversampling | 150, 197, 26, (6, 11), 0.14, 0.66, 0.07, 0.11, $2.6 \times 10^{-4}$, 0.98, ($-953.15$, 9148.94), $5.58 \times 10^{-6}$ |
| SMOTE | 108, 180, 17, (7, 9), 0.44, 0.03, 0.17, 0.35, $2 \times 10^{-4}$, 0.86, ($-557.91$, 103.76), $6.95 \times 10^{-5}$ |
| ADASYN | 157, 174, 18, (3, 8), 0.31, 0.33, 0.2, 0.15, $5.3 \times 10^{-5}$, 0.74, ($-566.18$, 9234.9), $4.4 \times 10^{-6}$ |
| Borderline SMOTE | 182, 167, 20, (7, 11), 0.17, 0.74, 0.015, 0.065, $4 \times 10^{-5}$, 0.62, ($-316.14$, 14.06), $2.57 \times 10^{-6}$ |
| Kmeans SMOTE | 103, 185, 17, (7, 9), 0.45, 0.36, 0.12, 0.064, $6 \times 10^{-4}$, 0.77, ($-687.65$, 8699.9), $3.32 \times 10^{-6}$ |

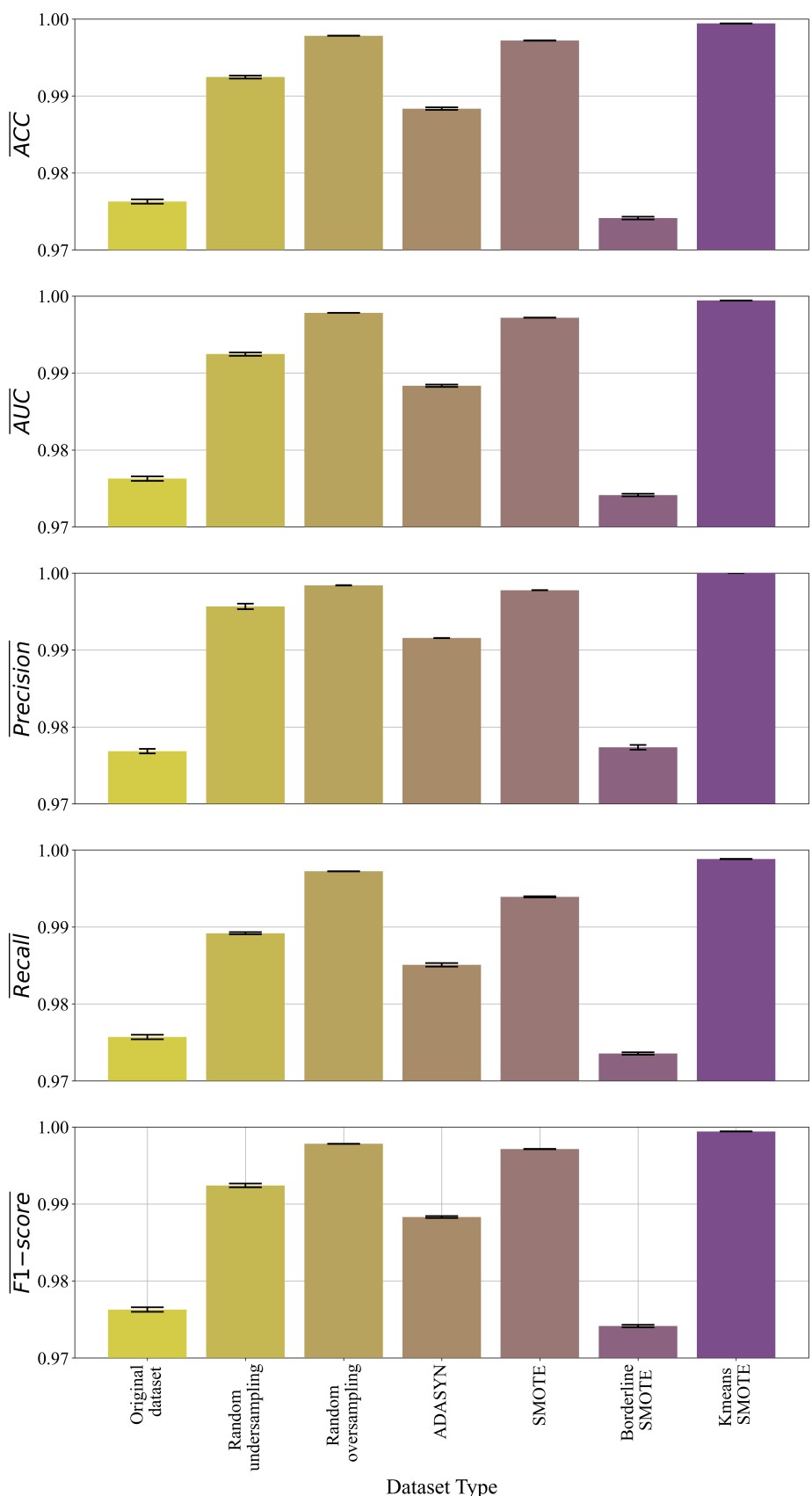

**Figure 7.** The mean and standard deviation of evaluation metric values achieved with GPSC, random hyperparameter search, and 5-fold cross-validation on each dataset variation.

As seen from Figure 7, the mean *ACC*, *AUC*, *Precision*, *Recall*, and *F1-Score* values are similar to those obtained in the previous investigation. However, the results obtained for all dataset variations showed a slight improvement in evaluation metric values. The only exception is in the case of symbolic expression obtained for the BorderlineSMOTE dataset which achieved lower evaluation metric values with the application of random hyperparameter search and 5-fold cross-validation than in the case of GPSC with random hyperparameter search method only. The *ACC*, *AUC*, *Precision*, *Recall*, and *F1-Score* values, in this case, dropped lower than in the case of the original dataset while the standard deviation value increased.

In Table 11 the mean and standard deviation values of *ACC*, *AUC*, *Precision*, *Recall*, and *F1-Score* are shown as well as average CPU time per simulation. The importance of this table was to primarily show how low the standard deviation values of used evaluation metrics are since they are not visible in Figure 7. The average CPU time per simulation, in this case, is much longer than in the previous case. This is because of the 5-fold cross-validation process, i.e., the training is performed on the training dataset 5 times on different folds. After the 5-fold cross-validation, the symbolic expressions are evaluated and if the values of all evaluation metrics are above 0.97 then the final train and test are performed. So in this case the GPSC is executed 6 times (5 for cross-validation and 1 train/test procedure).

**Table 11.** The mean and standard deviation of evaluation metric values used in GPSC with random hyperparameter search with 5-fold cross-validation.

| Dataset Type | $\overline{ACC}$ $pmSD(ACC)$ | $\overline{AUC}$ $\pm SD(AUC)$ | $\overline{Precision}$ $\pm SD(Precision)$ | $\overline{Recall}$ $\pm SD(Recall)$ | $\overline{F1\text{-}score}$ $\pm SD(F1\text{-}score)$ | Average CPU Time per Simulation [min] |
|---|---|---|---|---|---|---|
| Original Dataset | 0.976 $\pm 2.75 \times 10^{-4}$ | 0.97627 $\pm 2.75 \times 10^{-4}$ | 0.9768 $\pm 2.9 \times 10^{-4}$ | 0.9756 $\pm 2.8 \times 10^{-4}$ | 0.9762 $\pm 2.9 \times 10^{-4}$ | 320 |
| Random undersampling | 0.992 $\pm 1.9 \times 10^{-4}$ | 0.9924 $\pm 2.2 \times 10^{-4}$ | 0.9956 $\pm 3.6 \times 10^{-4}$ | 0.9891 $\pm 3.6 \times 10^{-4}$ | 0.9924 $\pm 2.5 \times 10^{-4}$ | 140 |
| Random oversampling | 0.997 $\pm 1.79 \times 10^{-5}$ | 0.9978 $\pm 4.11 \times 10^{-5}$ | 0.9983 $\pm 1.6 \times 10^{-5}$ | 0.9972 $\pm 7.9 \times 10^{-5}$ | 0.9978 $\pm 4.1 \times 10^{-5}$ | 400 |
| ADASYN | 0.9883 $\pm 1.71 \times 10^{-5}$ | 0.9883 $\pm 1.49 \times 10^{-5}$ | 0.9915 $\pm 8.26 \times 10^{-5}$ | 0.985 $\pm 2.2 \times 10^{-5}$ | 0.9883 $\pm 1.19 \times 10^{-5}$ | 400 |
| SMOTE | 0.9971 $\pm 2.47 \times 10^{-5}$ | 0.9971 $\pm 3.1 \times 10^{-5}$ | 0.9977 $\pm 9 \times 10^{-5}$ | 0.9939 $\pm 8.3 \times 10^{-5}$ | 0.9971 $\pm 2.6 \times 10^{-5}$ | 400 |
| Borderline SMOTE | 0.9741 $\pm 1.69 \times 10^{-5}$ | 0.9741 $\pm 1.6 \times 10^{-5}$ | 0.9773 $\pm 3 \times 10^{-4}$ | 0.9735 $\pm 1.56 \times 10^{-5}$ | 0.9741 $\pm 1.6 \times 10^{-5}$ | 400 |
| Kmeans SMOTE | 0.9994 $\pm 1.13 \times 10^{-5}$ | 0.9994 $\pm 1.2 \times 10^{-5}$ | 1.0 $\pm 0$ | 0.9988 $\pm 2.4 \times 10^{-5}$ | 0.9994 $\pm 1.2 \times 10^{-5}$ | 400 |

### 3.3. Best Symbolic Expressions

Based on obtained results in both cases the symbolic expressions with the best classification accuracy were obtained with the dataset that was balanced with the KMeansSMOTE method. The procedure necessary steps to use presented symbolic expression can be divided into the following steps:

- Data preparation—the procedure is described in the Materials and Methods section;
- Providing input variables to the symbolic expression and calculating numerical output;
- The obtained numerical output is provided as an argument of Sigmoid decision function (Equation (17)), and rounding the output of Sigmoid function to obtain 0/1 value.

In the case of GPSC with random hyperparameter search and train/test split in ratio 70:30 the best symbolic expression obtained on the dataset balanced with the KMeansSMOTE method can be written as:

$$
\begin{aligned}
y \;=\; & \left( \min(X_8, \log(X_2)) + |X_3| + X_8\left( \log\left( \frac{\cos(\cos(X_9))}{\sqrt{X_4}(X_7 - X_1)} \right) + \left( X_3 \right.\right.\right. \\[2mm]
+\; & (\log(X_7) + 6498.6)\left( \log\left( \cos\left( \frac{\log(X_5)}{\log(10)} \right) \frac{1}{\tan(\log(X_5))} \right) + \sqrt[3]{X_6} \right) - X_8 \right)^{\frac{1}{3}}\bigg)\bigg)^{\frac{1}{3}}.
\end{aligned}
\tag{22}
$$

As seen from Equation (22), the equation for detection of malicious websites consist of all input variables except the $X_0$ which is the **url_len** variable shown in Table 4. In the case of GPSC with random hyperparameter search with 5-fold cross-validation the best symbolic expression was also obtained on the dataset balanced with the KMeansSMOTE method and can be written as:

$$
\begin{aligned}
y \;=\; & \max\left( \sqrt[3]{X_6} - X_8, \max\left( \max\left( \sin(\tan(\tan(X_6))), \max\left( |X_3|, \right.\right.\right.\right. \\[2mm]
& \left( \max(\log_{10}(\max(\max(X_1 X_6, \max(X_3 - X_6, \max(X_6, \right. \\[2mm]
& \max(\max(\log_{10}(\log_{10}(\log_{10}(\log_{10}(\sqrt{\frac{\log_{10}(\frac{X_4}{\sqrt{X_3 - X_0}})}{X_6}} X_5)) X_8)), \\[2mm]
& \sqrt[3]{X_6}), \sqrt{\max(\log_{10}(\log_{10}(\log_{10}(\log_{10}(\sqrt{\frac{\log_{10}(X_6)}{X_6}} X_5)) X_8)), \sqrt[3]{X_6}) X_8)} \\[2mm]
& - \sqrt[3]{X_0})) - \sqrt{\log(X_8)}, \max(\tan(\tan(\log_{10}(X_6))), \sqrt[3]{X_9} - X_8)) X_8), \\[2mm]
& \tan(\log_{10}(X_6))) X_8)^{\frac{1}{2}})), X_4 \right) - \max(\log_{10}(\log(X_5)), \sqrt{X_0 X_4}) \right) - \sqrt{\frac{X_6}{X_6} X_8}.
\end{aligned}
\tag{23}
$$

As seen from Equation (23), the symbolic expression consists of all input variables except variables $X_2$ and $X_7$ and these input variables are **tld** and **net_type**, respectively. The **tld** represents a top-level domain while **net_type** represents the type of IP address. These two variables have extremely low correlation values with the target variable (tld with label = $-0.1$, net_type with label = 0) so it is obvious why these two variables were excluded during GPSC and did not end up in the symbolic expression.

### 4. Discussion

In the case of the classic train/test approach of the GPSC algorithm using seven different dataset variations the obtained $\overline{ACC}$, $\overline{AUC}$, $\overline{Precision}$, $\overline{Recall}$, and $\overline{F1\text{-}score}$ are all above 0.97 which is pretty high. Looking at Figure 2 it can be noticed that several variables, i.e., content_len, special_char, js_len, js_obf_len, HTTPS, and who_is variables have a high correlation (lowest negative correlation: $-0.7$, and highest positive correlation range: 0.7–0.9) to the target variable ("label"). So the high $\overline{ACC}$, $\overline{AUC}$, $\overline{Precision}$, $\overline{Recall}$, and $\overline{F1\text{-}score}$ of symbolic expression in the case of the original dataset can be attributed to the high correlation between several input variables and the desired output variable. Looking at Tables 9 and 11, the difference between train/test scores, i.e., standard deviation is the largest in the case of the original dataset. Regarding random oversampling and undersampling datasets, the $\overline{ACC}$, $\overline{AUC}$, $\overline{Precision}$, $\overline{Recall}$, and $\overline{F1\text{-}score}$ are almost the same. The only benefit of using random undersampling is that the dataset has a lower number of samples which can easily be used in training and testing the GPSC algorithm since it requires less memory.

From all other oversampling methods (ADASYN, SMOSTE, BorderlineSMOTE, and KmeansSMOTE) the highest result is achieved in the case of KmeansSMOTE and the lowest score in the case of the ADASYN dataset. Standard deviation between training/test in the

case for all symbolic expressions obtained with oversampling dataset methods is virtually nonexistent which means that scores between training and testing are almost the same. When compared to the standard deviation of the symbolic expression obtained on the original dataset it can be noticed that oversampling methods have had some influence on the classification accuracy of obtained symbolic expressions since these methods provided balanced datasets.

Regarding the hyperparameters used in the classic train/test approach, the crossover is not the dominant genetic operation which is usually the case. Looking at Table 8 for each dataset type the crossover was the dominant coefficient in 3 out of 7 cases. However, the sum of all genetic operations used to obtain the symbolic expression for each dataset is never equal to 1 which means that a small percentage of tournament winners entered the next generation unchanged during GPSC executions. The value of the parsimony coefficient is very small in all cases, i.e., $10^{-6}$ which could potentially indicate a bloat phenomenon. However, the bloat did not occur since the fitness value was constantly dropping but the size of the symbolic expressions increased which made longer execution times. The value of stopping_criteria was never reached, i.e., in the case of the classic train/test approach with a random hyperparameter search method. All GPSC executions were terminated after a predefined number of generations were reached. GPSC was unable to reach the predefined minimum fitness value (stopping_criteria value) although the obtained results (classification accuracies of obtained symbolic expressions) are exceptional.

In the case of GPSC with random hyperparameter search and 5-fold CV the list of hyperparameters is shown in Table 10 and results in Figure 7 and Table 11. The results in Figure 7 and Table 11 showed that the $\overline{ACC}$, $\overline{AUC}$, $\overline{Precision}$, $\overline{Recall}$, and $\overline{F1\text{-}score}$ are similar when compared to results shown in Figure 6 and Table 9. The $\overline{ACC}$, $\overline{AUC}$, $\overline{Precision}$, $\overline{Recall}$, and $\overline{F1\text{-}score}$ values achieved with a symbolic expression that was obtained using this method (GPSC with random hyperparameter search, and 5-fold cross-validation) on the original dataset are slightly higher than in the previous method (GPSC with random hyperparameter search on classic train/test). However, the values of standard deviation for all evaluation metric values are almost the same. Random oversampling, ADASYN and SMOTE produced almost the same results as in the previous case. The symbolic expressions obtained on the datasets balanced with random undersampling and BorderlineSMOTE methods achieved lower evaluation metric values when compared to the previous case. However, the highest classification accuracy was achieved again in the case of the dataset balanced with the KMeansSMOTE method with the lowest, almost nonexistent standard deviation.

From the hyperparameters shown in Table 10, the crossover coefficient is dominant only in two cases. However, the sum of all genetic operations in all cases is almost equal to 1 which again means that a small percentage of tournament winners entered the next generation unchanged. The parsimony coefficient value was again very low so the size of population members grew slightly during the execution which in the end prolonged the execution times. All the executions of GPSC with a random hyperparameter search method and 5-fold cross-validation were terminated after the maximum number of generations was reached which indicates that the randomly selected stopping criteria value (second termination criteria) was extremely low and was never reached by any of the population members during the GPSC execution.

When the results of other research papers summarized in Table 1 are compared with the results presented in this paper, it can be noticed that the values of evaluation metrics, i.e., accuracy, achieved in this research were even better than those presented in the table. This statement is valid for symbolic expressions obtained on datasets balanced with random undersampling, random oversampling, ADASYN, SMOTE, and KMeansSMOTE methods, respectively. The benefit of utilizing the GPSC algorithm when compared to the ML algorithms used in research papers presented in Table 1 is that after training a symbolic expression is obtained which requires less space for storage and faster detection than other ML algorithms. After training the ML algorithm if the learned algorithm is not stored

the ML algorithm has to be trained again. In the case of GPSC, after training with this algorithm, the symbolic expression is obtained and if the GPSC model is not stored the symbolic expression can be used for further investigation.

## 5. Conclusions

In this paper, the GPSC algorithm was applied to publicly available datasets to investigate if, using the aforementioned method, the symbolic expression could be obtained that can detect malicious websites with high classification accuracy. To improve classification accuracy, the GPSC was combined with a random hyperparameter search and 5-fold cross-validation. Additionally, different dataset balancing methods were investigated to see if the classification accuracy could be improved. Based on the conducted investigations, the following conclusions can be drawn, i.e.,

- The GPSC algorithm can be used to obtain symbolic expressions which could detect malicious websites with high classification accuracy;
- The application of GPSC with the random hyperparameter search method and 5-fold CV on various types of datasets achieved almost similar classification accuracy when compared to GPSC with the random hyperparameter search method;
- The application of oversampling methods showed that the samples of the minority class could be synthetically increased to balance the dataset classes and in the end improve classification accuracy. The KmeansSMOTE was the only dataset with which GPSC produced symbolic expression with high classification accuracy in both cases (with and without 5-fold CV).

The advantages of the proposed method are:

- After training using GPSC, the symbolic expression (mathematical equation) is obtained;
- Dataset undersampling and oversampling methods can improve the classification accuracy of the obtained symbolic expressions;
- 5-fold cross-validation process with random hyper-parameter search proved to be a powerful tool in generating symbolic expressions with high classification accuracy in the detection of malicious websites.

The disadvantages of proposed method are:

- The CPU time needed to train the GPSC algorithm depends on dataset size and GPSC hyperparameter values, i.e., larger datasets, population size, and number of generations could prolong the GPSC execution time;
- The parsimony coefficient is one of the most sensitive hyperparameters in GPSC. The range which will be used in the investigation had to be defined initially through trial and error. The low values can result in a bloat phenomenon while large values can choke the population and prevent the growth of each population member in size.

Future work regarding the detection of malicious websites will be focused on enlarging the dataset, especially the number of original malicious websites in the dataset. Another part of the future work will be focused on the improvement of classification accuracies by way of the implementation of dataset scaling/normalizing methods.

**Author Contributions:** Conceptualization, N.A., S.B.Š. and I.L.; methodology, S.B.Š. and M.G.; software, N.A. and I.L.; validation, N.A., I.L. and M.G.; formal analysis, N.A.; investigation, N.A. and I.L.; resources, S.B.Š. and M.G.; data curation, S.B.Š. and M.G.; writing—original draft preparation, N.A., S.B.Š. and I.L.; writing—review and editing, N.A., S.B.Š. and I.L.; visualization, N.A. and M.G.; supervision, N.A.; project administration, N.A.; funding acquisition, N.A. All authors have read and agreed to the published version of the manuscript.

**Funding:** This research received no external funding.

**Institutional Review Board Statement:** Not applicable.

**Informed Consent Statement:** Not applicable.

**Data Availability Statement:** Publicly available dataset was used in this study. This data can be found here: https://www.kaggle.com/datasets/aksingh2411/dataset-of-malicious-and-benign-webpages (accessed on 10 October 2022).

**Acknowledgments:** This research has been (partly) supported by the CEEPUS network CIII-HR-0108, European Regional Development Fund under the grant KK.01.1.1.01.0009 (DATACROSS), project CEKOM under the grant KK.01.2.2.03.0004, Erasmus+ project WICT under the grant 2021-1-HR01-KA220-HED-000031177, and University of Rijeka scientific grant uniri-tehnic-18-275-1447.

**Conflicts of Interest:** The authors declare no conflict of interest.

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
