# Peer review of "Detection of Malicious Websites Using Symbolic Classifier"

_futureinternet, doi:10.3390/fi14120358_

Round 1

Reviewer 1 Report

The paper presents a solution based on using a combination of hyper parameter tuning with GPSC algorithm. It is not clear why the authors decided to use GPSC algorithm and did not compare the efficacy of the algorithm against other solutions such as deep learning classifiers.  

Author Response

The authors want to thank the reviewer for his time, effort, and constructive suggestions that have greatly improved the quality of this manuscript. The authors of this manuscript hope that the changes made in this manuscript will provide a suitable scientific contribution.

The paper presents a solution based on using a combination of hyperparameter tuning with the GPSC algorithm. It is not clear why the authors decided to use the GPSC algorithm and did not compare the efficacy of the algorithm against other solutions such as deep learning classifiers. 

The Genetic programming Symbolic Classifier was used in this research since with this algorithm symbolic expressions can be obtained which can be used to detect malicious websites. Unlike deep learning methods or other machine learning methods, the authors of this manuscript wanted to show that by using the simple method GPSC symbolic expressions can be obtained. These symbolic expressions require less computational resources when compared to complex machine or deep learning classifiers in the detection of malicious websites. In other words, symbolic expression (mathematical equation) is more efficient than saved machine or deep learning classifier since it requires less memory space to store it and it is easier to load and process the symbolic expression than the trained machine or deep learning classifier. 

To improve the classification accuracy of generated symbolic expressions using GPSC the random hyperparameter search method as well as 5-fold cross-validation was developed and applied in this investigation.

However, the original dataset was imbalanced (a large number of benign websites and a small number of malicious websites) so one of the ideas was to investigate if balancing the dataset using different undersampling/oversampling methods could improve classification accuracy. 

In the revised version of the manuscript the Introduction section was divided into the following subsections (suggested by one of the reviewers) :

  • Blacklisting or heuristics approach, 
  • Machine learning algorithms, and
  • Definition of the idea, novelty, and research hypotheses

The idea and novelty are clearly stated in the third subsection of the novelty. Citing from the revised version of the manuscript (subsection “Definition of the idea, novelty, and research hypotheses): 

The idea of this paper is to obtain a symbolic expression (simple mathematical equation) that can detect malicious websites with high classification accuracy using genetic programming symbolic classifier (GPSC). The advantage of using this approach is that the result is a symbolic expression (mathematical equation) that requires less disk space for storage and is easier to use (faster investigation) when exploring a new website.

The novelty of this paper is to show the procedure of how GPSC can be utilized to obtain the symbolic expression for the detection of malicious websites. Besides the GPSC utilization, the dataset preparation procedure is presented, as well as dataset balancing methods since the original dataset has an imbalance between class samples. To obtain the symbolic expression with high classification accuracy the random hyperparameter search method and 5-fold cross-validation methods were developed and used in this research.

The third subsection was also added in the results section entitled: “Best symbolic expressions” in which the best symbolic expressions that were obtained in the case of dataset balanced with KMeansSMOTE method in both cases i.e. GPSC with random hyperparameter search and GPSC with random hyperparameter search and 5-fold cross-validation. 

The reason for showing the best symbolic expressions in terms of classification accuracy is that the idea was to obtain the symbolic expression which can be used for the detection of malicious websites.

Reviewer 2 Report

1.       The experimental results for case study in section 3 were provided without quality measures (standard error) in the current format. For example, the means of model accuracy in term of AUC were provided without standard deviation/error in Figures 6 and 7. Please add the corresponding standard error for the corresponding estimates in the revised manuscript.

2.       The authors should clarify that why the following performance measures, such as recall, precision, and F1 scores, were not considered in their work.

3.       The computational efforts (CPU time) did not provide in the manuscript. Especially the efforts spent on the training using different approaches.

4.       The author should clarify the pros and cons of the proposed method.

Author Response

The authors of the manuscript want to thank reviewer for his time and effort to give constructive comments and suggestions which could greatly improve the manuscript quality. The authors of the manuscript have made all necessary modifications to address the comments and suggestions made by the reviewer. 

  • The experimental results for case study in section 3 were provided without quality measures (standard error) in the current format. For example, the means of model accuracy in terms of AUC were provided without standard deviation/error in Figures 6 and 7. Please add the corresponding standard error for the corresponding estimates in the revised manuscript.

The authors want to suggest that reviewer 2 take a closer look at Figures 6 and 7 in the original version of the manuscript. The standard deviation is presented in form of the error bar in form of the letter “I”. However, due to small values of standard deviation/error, it is not so visible except in the case of “Original Dataset” and “Random undersampling” in Figure 6 and the case of “Original dataset” and “Borderline SMOTE” in Figure 7. Since the size of the error bars is too small the authors have provided additional tables with mean values of evaluation metrics used in the research and with standard deviation.  However, since this reviewer and other reviewers have raised the question of why recall, precision, and F1 scores were not considered they are now added to the modified version of the manuscript. (mean values obtained on train/Test dataset as well as standard deviation/error). Since the majority of described literature in the Introduction section used Accuracy as the evaluation method for ML algorithms this evaluation method was also used in the revised version of the manuscript for comparison with other research which is now described in the discussion section. 

 The results of all evaluation methods used in this research i.e. accuracy, AUC, precision, recall, and F1-score are graphically represented in figures 7 and 8 while numeric values are given in Tables 9 and 11. 

  • The authors should clarify why the following performance measures, such as recall, precision, and F1 scores, were not considered in their work.

The authors have used the AUC in the original investigation since they thought it was enough However, in the revised version of the manuscript the authors included the mean values of accuracy, AUC, precision, recall, and F1-Score  with standard deviation, and the metric was described in the sub-section Evaluation methodology which is now renamed to Evaluation Metrics and Methodology. 

 The results of all evaluation methods used in this research i.e. accuracy, AUC, precision, recall, and F1-score are graphically represented in figures 7 and 8 while numeric values are given in Tables 9 and 11. 

  • The computational efforts (CPU time) did not provide in the manuscript. Especially the efforts spent on the training using different approaches.

Alongside the results presented in Table 9 and Table 11 the average CPU time for each GPSC was provided for each dataset variation. Below the aforementioned tables, the average CPU time was described. The description of computational resources on which GPSC was executed is given in the Materials and methods subsection Computation Resources.

  • The author should clarify the pros and cons of the proposed method.

In the revised version of the manuscript, the pros and cons of the proposed method (GPSC) are given at the end of subsection GPSC.  The subsection entitled “The advantages and disadvantages of GPSC algorithm” 

The advantages and disadvantages of the proposed method are also given in at the end of the conclusion section. 

Reviewer 3 Report

The paper describes a Machine Learnig approach to find malicious websites. The research design is sound and the results are interesting. The detailed description of the oversampling and undersampling methods used is especially valuable (as in many other papers on ML where over/undersampling is used, this really crucial topic is often neglected). The paper is easy to read.

Some minor points need improvements:
a) In Table 1 the 9 sites that use https should show this in the URL-column, too for clarification.

b) On page 6 in line 203: "the constant variable..."; a variable is not constant by definition: either it is a constant or it is a variable.

c) In section 2.4.3 the entire dataset is denoted by "T" and the minority class as "S". In section 2.4.2, however, "S" is the name for the entire set of samples. It is not an error, but a bit confusing.

d) Table 5 does not contain very interesting information and could be discarded.

e) In section 2.5, line 333: "As seen from Figure 4 the max is a root node...". I think, that you should be min.

f) Some references are incomplete and the missing information must be added: 17, 23, 24, 28, 34

Author Response

The authors of the manuscript want to thank reviewer 3 for his valuable time and effort made to reviewing the manuscript and define comments and suggestions which could improve the manuscript's quality. The authors of the manuscript have corrected the manuscript according to comments and suggestions. The authors do hope that manuscript in this form will be accepted for publishing. 

The paper describes a Machine Learning approach to finding malicious websites. The research design is sound and the results are interesting. The detailed description of the oversampling and undersampling methods used is especially valuable (as in many other papers on ML where over/undersampling is used, this crucial topic is often neglected). The paper is easy to read.

Some minor points need improvements:

  • In Table 1 the 9 sites that use https should show this in the URL column, too for clarification.

In the revised version of the manuscript the URL column of Table 1 was modified i.e. the URLs were modified from http to https for those sites which had a “yes” value in the https column. 

  • On page 6 in line 203: "the constant variable..."; a variable is not constant by definition: either it is a constant or it is a variable.

The authors want to apologize for the mistake. Instead of “content” the authors have written a “constant” variable which makes no sense at all. The “content variable” is the column labeled “content” in Table 1 and of course in the dataset. In the modified version of the manuscript, the term “constant” was replaced with “content”.

  • In section 2.4.3 the entire dataset is denoted by "T" and the minority class as "S". In section 2.4.2, however, "S" is the name for the entire set of samples. It is not an error, but a bit confusing.

In section 2.4.2 the name for the entire set of samples was changed from S to T, the minority set of samples from S_min to T_min, and the majority set of samples from S_maj to T_maj.

  • Table 5 does not contain very interesting information and could be discarded.

Answer: Table 5 was omitted and instead a comment was given describing the number of samples for each method. Citing from the modified version of the manuscript: 

After application of ADASYN, SMOTE, BorderlineSMOTE, and KMeansSMOTE on the original dataset total of 4 new balanced data sets were obtained. With the application of these balancing methods, the balance in the number of samples of the majority and minority classes was equalized. In all four datasets, the total number of samples is  3053238, and the number of samples in class 0 and class 1 is equal to 1526619.

  • In section 2.5, line 333: "As seen from Figure 4 the max is a root node...". I think, that you should be min.

In the revised version of the manuscript the max was changed to min. 

  • Some references are incomplete and the missing information must be added: 17, 23, 24, 28, 34

In the revised version of the manuscript, the references were filled with additional information and are now completed. 

Reviewer 4 Report

I recommend a major revision based on the below points. Please, add a point-to-point response to each comment in your revision:

·        I am not convinced about the novelty of the manuscript.

·        The abstract is not technical and needs to highlight the research gap clearly.

·        The abstract also missed statistical information about the results.

·        The structure of the paper is vague. The paper needs to be restructured.

·        Don't add heading over heading. Add a few lines related to the detail of a particular section before starting a sub-section.

·        The novelty of the paper needs to be justified and clearly defined. It includes a clear difference between the available literature and previous works. The authors are asked to provide the limitations of the previous correlated works and then link those limitations to the current ideas and contributions of the current work.

·        Summarise the literature in the form of a table.

·        The heading should be literature review/related work.

·        Please avoid using the words "you," "we," or "our" in the manuscript. Please consider using phrases like "in this study/paper/Proposed/method" or another appropriate phrasing. This applies to the entire manuscript.

·        Add a Related Work section.

·        The overall approach/methodology is unclear.

·        The introduction section is quite extensive and needs to remove unnecessary details.

·        Proofread your paper from a native English speaker. There are many typos and grammar mistakes.

·        At the end of the Introduction section, add the contributions clearly. The mentioned contributions are not clear.

·        Related work/background/literature review should have a threat to a validity section. At the start of the background section, add a threat to a validity section. In that section, state the search strings and databases explored to find the related work.

·        The literature needs to be subdivided into multiple sub-sections.

·        Comparison with the state-of-the-art is missed. You need to compare your method with the ground truth.

·        It needs to add the reasons why these metrics are used for comparison.

·        Add the discussion related to the time complexity factor of AI models.

Overall, the paper has many inconsistencies, and the contributions are not clear. The results are not compared with the ground truth properly. Limitations are not provided in their current approach. Future directions are not clearly stated.

I am looking forward to seeing your revised version.

All the best.

Author Response

We want to thank the reviewer for his comments and suggestions. The majority of those comments were accepted and the manuscript was modified accordingly to improve its quality. However, the authors of this paper did deeper research into the literature suggested by the fourth reviewer. They found that in each of the 12 papers one author's name was constantly repeated. Due to the suspicion that all the articles that should be cited are the works of the fourth reviewer, the editor was contacted and the unethical behavior of the fourth reviewer in the review process was pointed out. One of the editor's suggestions was not to cite articles suggested by reviewer 4, so those comments were dropped from the review. The answers to the comments of 4th reviewer that were approved by the editor are given below. 

I recommend a major revision based on the below points. Please, add a point-to-point response to each comment in your revision:

  • I am not convinced about the novelty of the manuscript. 

The authors of the manuscript agree with this suggestion and have made an effort to clearly and explicitly as possible indicate the novelty of this paper. The novelty of this paper.  The novelty of this paper is to implement genetic programming symbolic classifier (GPSC) on a publicly available dataset to obtain symbolic expression (mathematical equation) which could be used for the detection of malicious websites. The novelty is mentioned throughout the manuscript. 

The first instance of paper novelty is mentioned in the abstract of the manuscript. Citing from the revised version of the manuscript ( abstract line): 

In this paper, a novel approach is proposed which consists of the implementation of Genetic programming symbolic classifier (GPSC) algorithm on a publicly available dataset to obtain simple symbolic expression (mathematical equation) which could detect malicious websites with high classification accuracy.

The second instance of the manuscript where the novelty is mentioned is the introduction. Now in the revised version of the manuscript, the novelty is located in the third (last) subsection of the Introduction section. Citing from the revised version of the manuscript: 

The novelty of this paper is to show the procedure of how GPSC can be utilized to obtain the symbolic expression for the detection of malicious websites. Besides the GPSC utilization, the dataset preparation procedure is presented, as well as dataset balancing methods since the original dataset has an imbalance between class samples. To obtain the symbolic expression with high classification accuracy the random hyperparameter search method and 5-fold cross-validation methods were developed and used in this research.

  • The abstract is not technical and needs to highlight the research gap. The abstract also missed statistical information about the results.

The abstract is modified and now contains very detailed information about the investigation conducted in this paper.  The revised version of the abstract now describes the novelty of this paper, how the investigations were conducted as well as the statistical results of the best symbolic expressions. Citing from the revised version of the manuscript: 

Malicious websites are web locations that attempt to install malware which is the general term for anything that will cause problems in computer operation, gather confidential information, or gain total control over the computer. In this paper, a novel approach is proposed which consists of the implementation of Genetic programming symbolic classifier (GPSC) algorithm on a publicly available dataset to obtain simple symbolic expression (mathematical equation) which could detect malicious websites with high classification accuracy. Due to a large imbalance of classes in the initial dataset several data sampling methods (random undersampling/oversampling, ADASYN, SMOTE, BorderlineSMOTE, and KmeansSMOTE) were used to balance the dataset classes. For this investigation, the hyperparameter search method was developed to find the combination of GPSC hyperparameters with which high classification accuracy could be achieved. The first investigation was conducted using GPSC with a random hyperparameter search method and each dataset variation was divided on a train and test dataset in a ratio of 70:30. To evaluate each symbolic expression the performance of each symbolic expression was measured on the train and test dataset and the mean and standard deviation values of accuracy (ACC), $AUC$, precision, recall and f1-score were obtained. The second investigation was conducted also using GPSC with random hyperparameter search method however, 70\% i.e. train dataset was used to perform 5-fold cross-validation. If the mean accuracy, $AUC$, precision, recall, and f1-score values were above 0.97 then final training and testing (train/test 70:30) were performed with GPSC with the same randomly chosen hyperparameters used in a 5-fold cross-validation process and final mean and standard deviation values of aforementioned evaluation methods were obtained.  In both investigations, the best symbolic expression was obtained in the case where the dataset balanced with the KMeansSMOTE method was used for training and testing. The best symbolic expression obtained using GPSC with random hyperparameter search method and classic train-test procedure (70:30) on a dataset balanced with KMeansSMOTE method achieved values of $\overline{ACC}$, $\overline{AUC}$, $\overline{Precsion}$, $\overline{Recall}$ and $\overline{F1-score}$ (with standard deviation) $0.9992 \pm 2.249\times 10^{-5}$, $0.9995 \pm 9.945 \times 10^{-6}$, $0.9995 \pm 1.09 \times 10^{-5}$, $0.999 \pm 5.17\times 10^{-5}$, $0.9992 \pm 5.17\times 10^{-6}$, respectively. The best symbolic expression obtained using GPSC with a random hyperparameter search method and 5-fold cross-validation on a dataset balanced with the KMeansSMOTE method achieved values of $\overline{ACC}$, $\overline{AUC}$, $\overline{Precsion}$, $\overline{Recall}$ and $\overline{F1-score}$ (with standard deviation) $0.9994 \pm 1.13\times 10^{-5}$, $0.9994 \pm 1.2\times 10^{-5}$, $1.0 \pm 0$, $0.9988 \pm 2.4\times 10^{-5}$,  and $0.9994 \pm 1.2\times 10^{-5}$, respectively.

  • The structure of the paper is vague. The paper needs to be restructured.

The initial structure of  the paper is divided into the following sections: 

  1. Introduction - provides literature overview, description of the idea and novelty, the definition of hypotheses, and a short description of the manuscript outline. 
  2. The materials and Methods - section is divided into the following subsections:
    1. Research Methodology - 
    2. Dataset description and preparation - 
      1. Dataset transformation - 
      2. Statistical data analysis - 
    3. Dataset balancing methods - 
      1. Random undersampling and oversampling methods 
    4. Oversampling methods 
      1. SMOTE
      2. Adasyn
      3. Borderline smote 
      4. KMeansSmote 
    5. GP: Symbolic Classifier
    6. Evaluation Methodology 
    7. Random Hyperparameter search 
    8. Cross-validation 
  3. Results - of conducted investigations are presented. The initial train/test results with random hyperparameter search, as well as 5-fold cross-validation with random hyperparameter search. 
    1. Result of GP Symbolic Classifier with random hyperparameter search 
    2. Results of GP symbolic classifier with random hyperparameter search and 5-fold cross-validation 
  4. Discussion - 
  5. Conclusions - 

In the revised version of the manuscript, the following modifications are made: Introduction section was modified i.e. divided into three subsections: Blacklisting or heuristics approach, Machine Learning Algorithms, and Definition of the research idea, novelty, research hypotheses, and scientific contributions. 

In the Materials and Methods section in the section of GPSC the advantages and disadvantages of GP are added (suggested by one of the reviewers). In this section, the Evaluation Methodology was extended since other evaluation metrics were used and the Computational Resources subsection was added to describe the hardware and software used in this research. 

In the results section, the Best symbolic expressions subsection is added to show the result of GPSC application to the problem,

  • Don't add heading over heading. Add a few lines related to the detail of a particular section before starting a sub-section.

The authors have searched throughout the manuscript and have not found the occurrence heading over heading. The instances where some lines are added between Section and sub-section are 

Section: Materials and Methods - Sub-Section: Research Methodology - In the original version of the manuscript one sentence separates the section and subs-section and the sentence is:  “In this section, the research methodology is presented as well as the used dataset (description and preparation), GPSC algorithm, random hyper-parameter search method, 5-fold cross-validation, and evaluation methodology.

In the revised version of the manuscript, the previous line was modified i.e. one missing sub-section name was added and this was the Dataset Balancing methods. The modified version of the sentence: In this section, the research methodology is presented as well as the used dataset (description and preparation), dataset balancing methods, GPSC algorithm, random hyper-parameter search method, 5-fold cross-validation, and evaluation methodology.”

Section: Results - sub-section: Results of GP symbolic classifier with random hyperparameter search - two sentences exist in the original version of the manuscript. Citing from the original version of the manuscript: “In this section, the results of GPSC applied on 7 different dataset variations including the original dataset are shown. The results are divided into two subsections i.e. the results obtained with GPSC with random hyperparameter search where each variation of the dataset was divided into train and test portions in ratio 70:30, and GPSC with random hyperparameter search and 5-fold cross-validation method applied on each dataset variation.

From the presented cases it can be concluded that few lines exist between sections and sub-sections so no additional modifications were made except for the sentence between the Materials and Methods section and the following sub-section in which one topic of this section was missing (Dataset balancing methods) since it was missing in the original manuscript version. 

  • The novelty of the paper needs to be justified and clearly defined. It includes a clear difference between the available literature and previous works. The authors are asked to provide the limitations of the previous correlated works and then link those limitations to the current ideas and contributions of the current work.

The authors agree with the reviewer's suggestion and have emphasized the novelty of this paper throughout the manuscript. 

  • In the revised version of the manuscript in the abstract, the novelty is clearly stated for the first time. 

Citing from the revised version of the manuscript: “In this paper, a novel approach is proposed which consists of the implementation of Genetic programming symbolic classifier (GPSC) algorithm on a publicly available dataset to obtain simple symbolic expressions (mathematical equations) which could detect malicious websites with high classification accuracy.
So the general novelty of this paper is to obtain a mathematical equation with GPSC which could be used to detect malicious websites. The equation is easier to use since it requires less space to store it and can be processed faster than a complex machine learning model or deep learning model. The aforementioned models require more memory and are generally harder to process. 

  • In the revised version of the manuscript in the Introduction section the novelty is clearly stated for the second time: 

Citing from the revised version of the manuscript “Introduction” section: “The novelty of this paper is to show the procedure of how GPSC can be utilized to obtain the symbolic expression for the detection of malicious websites. Besides the GPSC utilization, the dataset preparation procedure is presented, as well as dataset balancing methods since the original dataset has an imbalance between class samples. To obtain the symbolic expression with high classification accuracy the random hyperparameter search method and 5-fold cross-validation methods were developed and used in this research.

The novelty here is a little bit expanded when compared to the initial novelty shown in the abstract. 

Finally, due to other reviewer’s request, the authors have shown the best symbolic expression obtained in the case of GPSC with a random hyperparameter search method 

  • Summarise the literature in the form of a table.

All the relevant literature is summarized in Table 1, where the used methods are given as well as the achieved results. 

  • The heading should be literature review/related work.

All the literature relevant to this research was cited and described in the Introduction section i.e. in subsections: 

  • Blacklisting or heuristics approach
  • Machine Learning Algorithms

  • Please avoid using the words "you," "we," or "our" in the manuscript. Please consider using phrases like "in this study/paper/Proposed/method" or another appropriate phrasing. This applies to the entire manuscript.

The authors of the manuscript did not use the words “you”, “we” or “our” in the manuscript. To prove this the entire manuscript was searched and occurrences of these words were shown below. 

The original version of the manuscript was searched for the words “you”, “we”, and “our”. 

  • The word “you” was not found in the original version of the manuscript.
  • The word “we” was not found in the original version of the manuscript.
  • The word “our” was not found in the original version of the manuscript. 

As you can see the examples with “you”, “we” and “our” words do not exist since the authors did not use these words. However, the phrases “in this paper” and” proposed method” was used  

The occurrence of the “In this paper” phrase in the original version of the manuscript. 

  • Line 3: In this paper, a novel approach is proposed which consists of…
  • Line 103: “...with high classification accuracy the idea in this paper is to utilize genetic programming…”
  • Line 206: “In this paper, the undersampling, and oversampling methods were used to investigate their…” 
  • Line 288: “In this paper, the…” 
  • Line 326: “In this paper, all investigations with genetic programming symbolic classifier (GPSC),...”
  • Line 420: “However, in this paper, the score was not only shown in the case…” 
  • Line 450: “In this paper, a 5-fold cross-validation method was utilized to obtain symbolic expression-...” 
  • Line 508: “In this paper, the GP symbolic classifier algorithm was applied to publicly available…” 

The occurrence of the “proposed method” phrase in the original version of the manuscript: 

  • Line 94: “The proposed method …” 

  • Add a Related Work section.

In the revised version of the manuscript the related work section was emphasized in the introduction section  in the following subsections:

  • The overall approach/methodology is unclear.

Answer: The authors disagree with the comment since in the original and revised version of the manuscript in the last paragraph of the introduction section the outline of the manuscript is provided. In the first subsection of the Materials and Methods section Figure, 1 clearly shows the flowchart of research methodology i.e. flow of the investigation from the original dataset through undersampling/over-sampling methods, application of GPSC with random hyperparameter search, and GPSC with random hyper-parameter search with 5-fold cross-validation. 

The rest of the Materials and Methods section clearly describes the materials and used methods indicated in figure 1. First, the original dataset was described with the transformation of the dataset to a usable form. Then the statistics of the original dataset are shown. After that, the description of undersampling/oversampling methods used is given as well as the GPSC algorithm with random hyperparameter search and 5-fold cross-validation and evaluation methodology. 

  • The introduction section is quite extensive and needs to remove unnecessary details.

The authors of the manuscript have rewritten the entire Introduction section to answer other comments and suggestions made by this reviewer. The authors do hope that unnecessary details were removed. 

  • Proofread your paper from a native English speaker. There are many typos and grammar mistakes. 

The entire manuscript was proofread by a native English speaker, and the typos and grammar mistakes are corrected. 

  • At the end of the Introduction section, add the contributions. The mentioned contributions are not clear.

Answer: First of all in the original version of the manuscript at the end of the Introduction section the scientific hypotheses defined that were investigated throughout this manuscript. However, the scientific contribution is missing. So in a revised version of the manuscript, the scientific contribution is clearly stated. 

The scientific contributions are: 

  • investigate if GPSC can be applied to the dataset for the detection of malicious websites,
  • investigate if datasets balanced with undersampling and oversampling methods have any influence on classification accuracies of obtained symbolic expressions using the GPSC algorithm, 
  • investigates if the random hyperparameter search method, as well as 5-fold cross-validation, has any influence on the classification accuracy of obtained symbolic expressions in the detection of malicious websites.

  • Related work/background/literature review should have a threat to a validity section. At the start of the background section, add a threat to a validity section. In that section, state the search strings and databases explored to find the related work.

The related work/background/literature review is added in the introduction section and as suggested it was divided into multiple sections. The literature regarding the detection of malicious websites using machine learning methods has summarized in the table in the Introduction.  

  • The literature needs to be subdivided into multiple sub-sections.

The literature is subdivided into two subsections and these are:

  • Blacklisting or heuristics approach 
  • Machine Learning Algorithms
  • Comparison with the state-of-the-art is missed. You need to compare your method with the ground truth. 

In the discussion section, the comparison of results achieved in this research with other research is made. 

  • It needs to add the reasons why these metrics are used for comparison. 

Initially, the authors thought that the AUC score was enough for evaluation. However, other reviewers also suggested using other metrics (evaluation methods) so the authors decided to use accuracy, AUC, precision, recall, and F1-score. The authors have taken a closer look at other research papers that were cited in this manuscript and have discovered that the majority of them used accuracy metrics so for comparison we have used this method also. 

  • Add the discussion related to the time complexity factor of AI models. 

In the Materials and Methods, a subsection is added entitled description of used computational resources to describe the specification of the computer used in this research. In the results section, the average CPU time for obtaining symbolic expression in each case was shown and below these tables, the average CPU time is commented on as well in the discussion section.

Overall, the paper has many inconsistencies, and the contributions are not clear. The results are not compared with the ground truth properly. Limitations are not provided in their current approach. Future directions are not clearly stated.

I am looking forward to seeing your revised version. 

All the best.

Round 2

Reviewer 2 Report

The authors seem to have done their best to address the reviewers’ comments. I am recommending this paper for publication to acknowledge the very commendable attempts of the authors to modify and improve it.

Reviewer 4 Report

Congratulations